# Advancements in Image-Based Models for High-Grade Gliomas Might Be Accelerated

**DOI:** 10.3390/cancers16081566

**Published:** 2024-04-19

**Authors:** Guido Frosina

**Affiliations:** Mutagenesis & Cancer Prevention Unit, IRCCS Ospedale Policlinico San Martino, Largo Rosanna Benzi 10, 16132 Genova, Italy; guido.frosina@hsanmartino.it

**Keywords:** image-based models, radiobiological prediction, artificial intelligence, precision radiation oncology, radiomics, radio genomics, machine learning, end of life

## Abstract

**Simple Summary:**

We review recent advances in imaging techniques and applications of artificial intelligence to improve the diagnosis and prognosis of high-grade gliomas, the deadliest brain tumors. Although these technological advances promise to improve the precision of radiotherapy and optimize treatment, they have yet to translate into widespread clinical benefits for patients with high-grade glioma. We discuss possible measures to accelerate technology transfer from bench to bedside. High-grade gliomas, to date, remain essentially fatal tumors, and the often unmet need to adapt legislative instruments to the end of life of patients is also discussed.

**Abstract:**

The first half of 2022 saw the publication of several major research advances in image-based models and artificial intelligence applications to optimize treatment strategies for high-grade gliomas, the deadliest brain tumors. We review them and discuss the barriers that delay their entry into clinical practice; particularly, the small sample size and the heterogeneity of the study designs and methodologies used. We will also write about the poor and late palliation that patients suffering from high-grade glioma can count on at the end of life, as well as the current legislative instruments, with particular reference to Italy. We suggest measures to accelerate the gradual progress in image-based models and end of life care for patients with high-grade glioma.

## 1. Introduction

High-grade gliomas (HGG; WHO grades III and IV) are brain tumors with the highest incidence and mortality. Fears of an increase in their incidence periodically emerge about the widespread and intense use of mobile phones, but the current epidemiological evidence shows no increased risks in this sense, even if the existence of several confounding factors makes the conclusions non-definitive [1,2]. Exposure to ionizing radiation, therefore, remains the only demonstrated risk factor for brain tumors [3,4].

With regard to mortality, HGGs remain extremely difficult to treat, with the last significant therapeutic advance dating back 18 years with the introduction of the alkylating agent temozolomide (TMZ) [5]. HGGs are intensively studied for their genetic and epigenetic characteristics [e.g., the methylation of the O^6^ methylguanine DNA methyltransferase (MGMT) promoter, mutations in the isocitrate dehydrogenase (IDH) gene or histone genes] as well as alterations of some signaling pathways that regulate cell proliferation (e.g., PI3K/AKT/mTOR, ATM, miRNA, WNT/β-catenin, JAK/STAT and others) in order to find more specific and personalized treatments [6,7,8]. However, the great heterogeneity, infiltrative capacity, and a plethora of alternative resistance pathways make the task extremely difficult in clinical settings [9]. Recently, alterations in the MAPK pathway such as BRAF V600E mutations have attracted attention in this context for the efficacy of the specific inhibitors of this altered signaling pathway in the therapy of some pediatric, and potentially adult, gliomas [10,11,12].

Unlike the problematic research into new therapeutic approaches, the development of image-based models for the diagnosis and prognosis of HGG is progressing more steadily, as evidenced by the regular publication of successive and increasingly complex versions of the classification of brain tumors, the most recent of which dates back to 2021 [13]. However, some measures could be taken to accelerate the development and validation of increasingly less invasive procedures. We discuss the topic after examining the most relevant advances published in the first half of 2022. In particular, magnetic resonance imaging (MRI), positron emission tomography (PET) and other diagnostic models developed in both the preclinical and clinical setting will be reviewed. Their employment for differentiating between HGG and specific pathologies or dealing with specific aspects (e.g., the diagnosis of pediatric HGG) will be analyzed. Advances in prognostic models developed preclinically and clinically will follow. Eventually, we will discuss some aspects related to the end of life of HGG patients. 

## 2. Materials and Methods

A literature search was performed in PubMed using general terms (“radio* and gliom*”) and time limits from 1 January 2022 to 30 June 2022. Despite the fact that the exclusion of relevant articles retrievable in other databases or using different terminology cannot be excluded, the use of the general terms indicated above has shown, in the past, to recover the vast majority of articles relating to the topic analyzed [14,15]. Of the 934 records automatically identified, 633 were manually discarded because, based on the article titles, they did not meet the criteria for relevance to the topics “Radiology of HGG” or “Radiotherapy of HGG”. In case of doubt, reference was made to the content of the abstract. Among the remaining 301 bibliographic references, another 79 were deemed ineligible based on the content of the article abstracts. In case of doubt, reference was made to the content of the entire article. The .pdf files of the 222 remaining articles were all retrieved and deemed suitable based on their content. Ninety were further analyzed and manually assigned to the “HGG Radiotherapy” group (discussed elsewhere) while the other 132 were assigned to the “HGG Radiology” group and discussed here. In addition to the 132 articles published in the first half of 2022, this review includes 38 articles published before 1 January 2022 or after 30 June 2022, due to their important relevance to the topic addressed or the topic eventually discussed on the end of life of patients, for a total of 170 references. A relevant flowchart is shown in Figure 1.

## 3. Models for the Diagnosis of HGG

### 3.1. MRI

#### 3.1.1. Preclinical Studies

HGG and brain abscesses may exhibit similar features on MRI imaging, such as expandable masses delimited by a ring with high contrast. Using animal models of dogs and cats, Carloni and collaborators investigated whether there are MRI imaging characteristics that might differentiate the two types of lesions [17]. Homogeneous signal intensity was frequent in intraaxial abscesses while a progression of signal intensity in the center of the lesion was indicative of glioma.

Ebrahimpour et al. elaborated an early detection method of HGG in animal models using a sequential administration of 5-aminolevulinic acid (ALA) and iron supplements [e.g., ferric ammonium citrate (FAC)] [18]. This sequential administration allowed an increase in iron deposition in C6 tumors transplanted into Wistar rats, allowing their better visualization by 3T MRI.

#### 3.1.2. Clinical Studies

Once the diagnosis is performed, surgical, chemotherapy and RT treatments are usually monitored using MRI sequences. Sometimes, MRI is unable to distinguish between tumor tissue, pseudoprogression phenomena and treatment damage, for the distinction of which we use perfusion techniques to evaluate the intensity of blood flow in the lesion possibly dependent on tumor neoangiogenesis or visualization via tomography with positron emission (PET), aimed at determining the metabolic activity of the lesion. These advances in imaging have partially extended to the operating room, making surgical resections more precise and safer. However, despite the accumulated experience, it can very often still be difficult for the neuro-radiologist to distinguish between tumor types using MRI [e.g., HGG or primary central nervous system lymphoma (PCNSL)], pseudoprogression and treatment necrosis [19]. 

Radiomic techniques for the extraction and digital quantification of data that may support the experience of the neuro-radiologist with the power of artificial intelligence (AI) are under development [20]; but they are still far from widespread use in clinical practice due to the heterogeneity of protocols and methodologies adopted in the different clinical trials in which machine learning techniques were used for the classification, diagnosis, prognosis and response to the treatment of HGG [21,22].

The relative cerebral blood volume (rCBV) is an important parameter of the vascularization and neoangiogenesis of the tumor lesion under investigation. It is usually quantified by T2-Perfusion Weighted Imaging (T2-PWI; Table 1). Seo and coworkers reported that, alternatively, T1-Perfusion Weighted Imaging with high temporal resolution can also provide reliable rCBV values [23]. 

Scola et al. compared dynamic magnetic resonance susceptibility contrast (DSC-PWI) perfusion techniques and computed tomography perfusion (CTP) in determining the vascularity of primary and secondary brain tumors. Perfusion associated with computed tomography (CT) has proved to be a valid alternative, should perfusion with magnetic resonance imaging (MRI) not be available or tolerated by the patient [24]. 

Stadlbauer and collaborators investigated whether machine learning applied to physiological MRI analysis (so-called “radiophysiomics”) could help to correctly diagnose brain tumors with contrast enhancement (CE) and different histology (glioblastoma, anaplastic astrocytoma, meningioma, PCNSL, brain metastasis) in comparison to conventional MRI (cMRI) or advanced (advMRI) techniques [25] (Figure 2A) (Table 1). PhyMRI has allowed the quantification of a series of diagnostic parameters not obtainable with cMRI or adv MRI such as microvascular architecture, neovascularization, oxygen metabolism and tissue hypoxia, thus contributing to the differential diagnosis of lesions. However, the large amount of time and effort dedicated to preprocessing data often requires deep neural networks.

The new NeuroXAI software (v.1) was applied in the classification and segmentation of the MRI images of brain tumors [26] (Figure 2B–H) (Table 1). Due to its public accessibility, ease of use and upgradability, NeuroXAI could be explored to assist in the neuroradiological diagnosis of brain tumors. This software is currently freely accessible at https://github.com/razeineldin/NeuroXAI (accessed on 17 April 2024). 

Lerner and collaborators conducted both a prospective and retrospective study where they suggested that the dosimetry and patient positioning for the RT treatment of HGGs can be performed exclusively with MRI, without resorting to CT [27] (Figure 2I,J) (Table 1). 

Ammari and colleagues proposed a machine learning method useful to reduce the amount of contrast medium in MRI (Table 1). The study demonstrated that, for lesions larger than 10 mm, it was possible to reduce, by 4 times, the contrast medium used, without significant loss of resolution. This could reduce exposure to gadolinium and its possible side effects in some patients. However, the conclusions were not extendable to smaller lesions, for which the standard amount of the contrast medium could only be reduced by compromising the imaging analysis quality [28].

**Figure 2 cancers-16-01566-f002:**
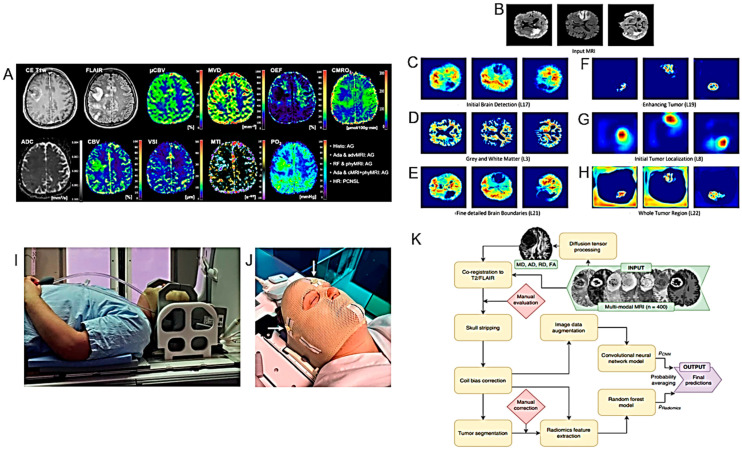
Models for the diagnosis of HGG I. (**A**). Representative case of a patient suffering from an anaplastic glioma (AG, WHO grade 3) who was misclassified as primary CNS lymphoma (PCNSL) by the radiologists but correctly classified by all three best-performing machine learning classifiers. CE T1w and FLAIR MRI data were conventional MRI (cMRI) data; cMRI data combined with the quantitative maps of ADC and cerebral blood volume (CBV) were advanced MRI (advMRI) data; physiological MRI (phyMRI) data included the quantitative maps of microvascular cerebral blood volume (µCBV), microvessel density (MVD) microvessel radius (VSI), microvessel type indicator (MTI), oxygen extraction fraction (OEF), cerebral metabolic rate of oxygen (CMRO2), and tissue oxygen tension (PO2), respectively. (**B**–**H**). Visualization of the information flow in the segmentation CNN internal layers. The input MRI sequences are shown in (**B**). (**C**–**E**) show implicit concepts for which no ground truth labels are available, in addition to explicit concepts (**F**–**H**) with trained labels. L stands for convolutional layer. (**I**). RT-setup of patient in three-point fixation mask scanned on a flat tabletop with 6-channel receiver flex coils (left and right) combined with an 8-channel posterior array (under the flat tabletop). (**J**). Fixation mask with liquid markers front, left and right, indicated by the white arrows. (**K**). Flow chart of MRI processing by combining radiomics and deep CNN features for predicting clinically relevant genetic biomarkers in GB. MR images (INPUT) were passed through several sequential automated image processing steps (rounded boxes) with minimal human input (diamond boxes). Each limb of the prediction model (CNN and radiomics) yields a single output logit, which is transformed into a probability using the sigmoid function. Final predictions (OUTPUT) were generated by averaging probabilities from the 2 limbs. Diffusion-tensor-derived contrasts include the following: mean diffusivity (MD), axial diffusivity (AD), radial diffusivity (RD), and fractional anisotropy (FA) (after [25,26,27,29] with permission).

A new non-invasive imaging methodology, in which radiomics features, convolutional neural network (CNN) features and a combination of both are used, was developed by Calabrese and collaborators to investigate some genetic biomarkers in patients with GB (WHO grade 4 diffuse astrocytic gliomas) [29] (Figure 2K) (Table 1). Comparison with laboratory genetic tests suggested that combining radiomic features and CNN obtained by MRI could improve the prediction of preoperative genetic markers.

Duran-Peña et al. have proposed a methodology for diagnosing HGG of the brainstem where the execution of a biopsy involves multiple and serious risks since it is the anatomical structure most involved in vital functions. They provided an algorithm illustrating the diagnostic process to limit the use of biopsy in this area to only essential and reasonably practicable cases [30]. 

**Table 1 cancers-16-01566-t001:** Models for the diagnosis of HGG.

Major Finding	Experimental System	Ref.
*MRI*		
T1-relative CBV effectively diagnosed progressive lesions in patients with HGG, suggesting the potential role of T1-PWI as a valid alternative to the traditional T2*-PWI.	45 MRIs of 34 patients with proved HGG.	[23]
Machine learning-based radiophysiomics might contribute to the clinical diagnosis of CE brain tumors.	A training cohort of 167 patients suffering from one of the five most common brain tumor entities (glioblastoma, anaplastic glioma, meningioma, primary CNS lymphoma, or brain metastasis).	[25]
The NeuroXAI software might be helpful in the detection and diagnosis of BTs.	NeuroXAI framework offering state-of-the-art XAI methods for classification and segmentation for both 2D and 3D medical image data.	[26]
Validation of MRI-only brain RT by a prospective clinical trial.	21 glioma patients.	[27]
Proposal of a deep learning method for virtual CE T1 brain MRI prediction.	200 multiparametric brain MRIs of a total of 145 patients.	[28]
A combination of radiomics and CNN features might improve the prediction performance of noninvasive genetic biomarkers.	Preoperative MRI data from 400 patients with GB who underwent resection and genetic testing.	[29]
Proposal of an an automated method to quantify the subtle deformations that occur in the peritumoral regions.	229 MRI exams from 27 patients with histologically confirmed HGG.	[31]
APTw MRI imaging shows good scan–rescan reproducibility in healthy tissue and tumors.	21 healthy volunteers and 6 glioma patients (4 GBs, 1 oligodendroglioma, 1 radiologically suspected LGG).	[32]
APTw MRI max values correlate positively with rCBVmax.	40 adult patients, treated for histopathologically confirmed glioma (WHO grades II–IV).	[33]
APTw MRI mean values might be helpful in the differential diagnosis of HGGs and meningiomas or HGGs and LGGs.	Imaging data of 50 BTs confirmed by pathology.	[34]
The metabolite ratios and the results of glioma grading obtained by MRS are affected by the image quality.	98 glioma patients confirmed by pathology.	[35]
ASL and DSC have similar diagnostic accuracy.	115 BT patients who underwent both ASL and DSC perfusion in the same 3T MRI scanning session.	[36]
Semi-quantitative analysis using SWI may contribute to the differential diagnosis between HGG recurrence and radionecrosis, but cannot identify BM.	56 patients with BM and 42 patients with HGG.	[37]
Study of an analytical qualitative algorithm to differentiate HGG from BM.	36 patients with histologically proven HGG or solitary BM matched by size and location.	[38]
Study of ^1^H-MRS to differentiate primary and secondary brain neoplasms.	61 MRI and ^1^H-MRS images of patients with histologically confirmed BTs.	[39]
CCA might help to distinguish PsP or RN from PD after RT.	16 patients with a primary and 17 with a secondary BT.	[40]
SWI permits to identify haemorrhagic changes due to anti-VEGF drugs.	A case of pseudoprogression after ioRT and regorafenib therapy in a patient with anaplastic astrocytoma recurrence.	[41]
A radiomics approach useful to predict pseudoprogression.	131 patients with HGG.	[42]
Use of the tissue permeability and microcirculation parameters Ktrans, Kep, IAUC to differentiate PT from TM.	34 patients with HGG.	[43]
Features of conventional MRI and RT treatment such as radiation dose, marginal enhancement and isointense ADC-signal may be useful to distinguish between progressive disease and TIE.	HGG adult patients who were treated with chemo/RT and subsequently developed a new or increasing CE lesion on conventional follow-up MRI.	[44]
Unlike the quantitative measurements of DSC and DCE perfusion maps, their qualitative assessment has low inter-examinator agreement.	HGG patients who underwent re-resection of a new enhancing lesion on post-treatment 3T MR examination including DWI, DCE and DSC sequences.	[45]
Smaller than reported longitudinal changes in MD, FA, and RD after VMAT or tomotherapy RT treatment.	27 patients newly diagnosed with GB and planned for VMAT or tomotherapy.	[46]
No significant improvement in therapy of DIPG in last decades. Post-RT necrosis is a frequent serious problem.	Medical records of 162 DIPG patients who underwent RT as an initial treatment.	[47]
LTS DIPG patients older at presentation compared to STS. *ATRX* mutation rates higher in this population than in the general DIPG population.	152 patients ≥10 years of age at diagnosis with imaging confirmed DIPG.	[48]
Importance of central neuro-imaging review in the diagnosis of DIPG.	Cases submitted to the International DIPG registry (IDIPGR) with histopathologic and/or radiologic data.	[49]
Radiomics as prognostic tool to stratify DIPG patients.	89 DIPG patients.	[50]
Variability of thalamic involvement of DMGs and its poor prognosis irrespective of *H3 K27* subtype alterations.	42 patients with radiologically evaluable thalamic-based DMG.	[51]
Quantitative ^23^Na MRI values in pediatric gliomas are higher than in normal tissues.	26 pediatric patients with gliomas scanned with ^23^Na MRI.	[52]
PsP is frequent after RT of pediatric LGG and independent of the RT modality (IS vs. XRT vs. PBT).	Baseline and follow-up MRI of 136 LGG patients.	[53]
A non-GBCM-enhanced protocol was non-inferior to a GBCM-enhanced protocol for follow up of OPGs.	42 children with isolated OPG.	[54]
After DKI of the peritumoral edema area, significant differences between grade III and IV gliomas. DKI parameters correlate with Ki-67.	51 patients with gliomas undergoing DKI scans before surgery.	[55]
A machine learning model predicting the *IDH* mutation status of gliomas.	69 patients with treatment-naïve diffuse glioma scanned with CEST MRI, DWI, FLAIR, and CE T1-weighted imaging at 3 T.	[56]
A radiomics model based on DCE-MRI and DWI predicting the *IDH1* mutation and angiogenesis in gliomas.	100 glioma patients examined with DCE-MRI and DWI.	[57]
The rCBV and PSR from DSC-MRI may predict the *IDH* mutation status in HGGs.	58 patients with histopathologically proved HGGs.	[58]
Asynchrony in vascular dynamics determined by resting-state BOLD fMRI, correlates with tumor burden and permits to delineate tumor boundaries in *IDH*-mutated gliomas	10 treatment-naïve patients with *IDH*-mutated gliomas who received standard-of-care preoperative imaging as well as echo-planar resting-state BOLD fMRI.	[59]
The pH- and oxygen-sensitive MRI is a feasible imaging technique for distinguishing glioma subtypes and determining their prognosis	159 adult glioma patients scanned with pH- and oxygen-sensitive MRI at 3T.	[60]
Quantitative relaxometry using syMRI may differentiate astrocytomas from oligodendrogliomas with increased sensitivity and objectivity compared to T2-FLAIR	13 patients with *IDH*-mutant diffuse gliomas, including 7 with astrocytomas and 6 with oligodendrogliomas.	[61]
The number of tumor blood vessels permits differentiating *IDH1* mutations	44 glioma patients [16 with *IDH1* mutant-type (*IDH1*-MT), 28 with *IDH1* wild-type (*IDH1*-WT)].	[62]
DWI and PWI MRI features may help to predict the *H3 K27M* mutation status in DMGs	94 DMG cases (mDMG = 48 and WT-DMG = 46).	[63]
The multiparametric MRI-based radiomic models may help to predict the *H3 K27M* mutation status in DMG	102 patients with pathologically confirmed DMG (27 with *H3 K27M*-mutant and 75 with *H3* WT status).	[64]
*PET*		
^18^F-DPA-714 and ^18^F-FDOPA correlate with the *IDH1* mutation in HGG.	U87 human GB isogenic cell lines with or without the *IDH1* mutation grafted into rat brains, and examined, in vitro, in vivo and ex vivo. PET imaging sessions, with radiotracers specific for glycolytic (^18^F-FDG), amino acid (^18^F-FDopa) and inflammation metabolism (^18^F-DPA-714).	[65]
^68^Ga-DOTA-(Ser)3-LTVSPWY specifically recognizes HER2 receptors.	U87 GB cell line and xenografted U87 GB tumor-bearing mice.	[66]
Multiparametric ^18^F-FDG PET/MRI diagnostic model based on conventional MRI features and quantitative analysis of the enhancing tumors and peritumoral regions is superior to single parameter in the differentiation of HGG and PCNSL.	45 patients with HGG and 20 patients with PCNSL undergoing simultaneous ^18^F-FDG PET, ASL PWI and DWI with hybrid PET/MRI before treatment.	[67]
^18^F-FET PET can avoid the negative consequences of premature chemotherapy discontinuation.	Effectiveness and cost effectiveness of serial ^18^F-FET PET imaging determination by analysis of published clinical data.	[68]
^18^F-FDOPA PET may contribute to prediction of glioma molecular parameters.	72 retrospectively selected, newly diagnosed glioma patients with ^18^F-FDOPA PET dynamic acquisitions.	[69]
Evaluation of ^18^F-DOPA PET-guided re-irradiation for progressive HGG.	20 adults with recurrent or progressive HGG previously treated with RT.	[70]
Elevated FBY activity found in primary GB, recurrent glioma and metastatic brain tumor which may suggest boron neutron capture therapy.	35 patients with 36 lesions prospectively examined with FBY PET and MRI.	[71]
PSMA expression evaluated prospectively in recurrent HGG using Glu-NH-CO-NH-Lys-(Ahx)-[^68^Ga-68 (HBED-CC)]-(^68^Ga-68 PSMA) PET.	49 lesions from 30 patients detected on MRI and further analyzed by fused PET/MRI with ^68^Ga-PSMA.	[72]
7T MRS compared to PET. Gln and Gly suggested as possible PET tracers.	In 24 HGG patients, 7T MRS and routine PET were co-registered and hotspot volumes of interest (VOI) were compared.	[73]
*Other diagnostic models*		
Elevated NADH is a metabolic consequence of TERT expression in cancer. [U-^2^H]-pyruvate is related to early response to therapy, prior to anatomic modifications.	RNAi, doxycycline-inducible expression systems, and pharmacologic inhibitors used in preclinical patient-derived tumor models.	[74]
Routine use of genomic and/or epigenomic profiling proposed to accurately classify gliomas.	38 adult patients with *IDH*-wild-type diffuse astrocytic gliomas lacking necrosis or microvascular proliferation on histologic examination.	[75]
Hypothesis that molGBs are histological GBs diagnosed early.	65 patients diagnosed with molGB.	[76]
Patients with DAG g experience clinical courses similar to GB.	25 patients matching the DAG g diagnosis.	[77]
Proposed use of molecular profiling to guide enrolment in early phase trials.	Patients enrolled in early phase trials of cytotoxic therapies, small molecule inhibitors or monoclonal antibodies from 2008 to 2018.	[78]
The CT-based TA may help in differentiating between primary and secondary malignancies.	36 patients with solitary BTs examined by CT.	[79]
Intramitochondrial heme biosynthesis factors as pharmacological targets to enhance intraoperative 5-ALA fluorescence visualization.	19 strongly fluorescing and 21 non-fluorescing tissue samples from neurosurgical adult-type diffuse gliomas (WHO grades II–IV).	[80]
Grade-specific cerebrovascular dysregulation in the entire brain of glioma patients.	96 patients with histologically confirmed cerebral glioma.	[81]
Systemic inflammatory biomarkers may contribute to the differential diagnosis of PCNSL from HGG.	42 PCNSL versus 16 HGG patients.	[82]
Gyriform infiltration is a specific imaging marker of molecular GBs.	426 patients: 31 molecular GB, 294 *IDH*-wild-type GB, 50 *IDH*-mutant astrocytoma, and 51 *IDH*-mutant 1p19q-codeleted oligodendroglioma.	[83]
Molecular investigations play an important role in the diagnosis and therapy of iHGG.	11 children under five years of age with newly diagnosed HGG.	[84]
A specific diagnostic pathway proposed for patients with suspected TDL.	41 TDLs and 91 HGG patients.	[85]

The results of the first MRI performed during the first six months after the end of chemo/radiotherapy (RT) have an important prognostic role, especially if it shows the compression of tissues adjacent to the surgical site [86]. The compression of healthy peritumoral tissue in brain tumor patients is considered a major cause of life-threatening neurological symptoms. After examining longitudinal MRI studies of patients with HGG, Fuster-Garcia et al. published a methodology for quantifying subtle peritumoral deformations and relating them to patient overall survival (OS) (Figure 3A,B) (Table 1) [31]. As expected, lower tumor compression on surrounding tissues was associated with greater patient survival.

Amide proton transfer (APT) imaging is a novel MRI technique that detects proteins and peptides in tissue via the saturation of the amide protons in the peptide bonds [87]. It may offer several potential clinical applications, but its reproducibility has to be adequately studied. Warmelink and colleagues reported that APTw MRI indeed exhibits good scan–rescan reproducibility in healthy tissue and HGG tumors with clinically usable scan times at 3 T [32] (Figure 3C–F) (Table 1).

The relationship between APTw MRI and DSC perfusion was investigated by Friismose and colleagues [33] who evaluated whether APTw MRI could be used as an alternative to magnetic resonance DSC in the imaging of HGG in adult patients (Figure 3G–M) (Table 1). APTw MRI (max) values correlated positively with rCBV (max) in patients treated for cerebral glioma. APTw MRI was confirmed as a reproducible technique but with some observer dependence. Although the results need to be confirmed by a larger population analysis, APTw MRI might usefully contribute to the follow-up imaging of HGG and represent a potential alternative to perfusion, especially in patients in whom the contrast medium is contraindicated. 

Zhang and co-workers used APTw MRI for the differential imaging between HGG, low-grade gliomas (LGG) and meningiomas and the determination of their invasiveness [34] (Figure 3N–R) (Table 1). Mean APT values could be used for the differential diagnosis of HGG and meningiomas or HGG and LGG. Using APTw MRI, gliomas showed more marked imaging features than meningiomas, possibly due to their higher infiltrative capacity.

7T MRS may be more sensitive to aberrant metabolic activity than lower-field strength MRS [88]. However, the 7T is not widely used in the clinics, and technical improvements are needed. Image resolution was confirmed as essential for the correct diagnosis of HGG by MRS in a study conducted by Shakir and collaborators [35] (Table 1). Metabolite ratios may be altered depending on the resolution and compromise diagnostic accuracy. The metabolic ratio that most effectively allowed HGG to be distinguished from LGG was that between choline and creatine (Cho/Cre).

Lavrova et al. compared the arterial spin labelling (ASL) MRI technique with DSC perfusion for the follow-up of primary and metastatic brain tumors at 3T, both in terms of lesion perfusion parameters and diagnostic accuracy [36] (Figure 4A–D) (Table 1). ASL and DSC were found to have roughly the same diagnostic accuracy, suggesting that ASL could be used as an alternative to DSC to measure perfusion in HGG and brain metastases. 

#### 3.1.3. Differentiating between HGG and Specific Pathologies

##### Lymphoma and other Primary Tumors

Distinguishing HGGs from PCNSL represents a diagnostic challenge with important therapeutic implications. Biopsy is still the preferred diagnostic method, but MRI, possibly reinforced by machine learning techniques, also contributes significantly to the differential diagnosis. AI methodologies have shown potential in the differential diagnosis between HGG and PCNSL, but such methodologies still lack large and balanced datasets and external validation. Cassinelli and coworkers highlighted the multiple deficiencies in the quality of the reporting and the risk of bias, which can reduce the generalizability and reproducibility of the results [89]. 

Astroblastoma is a rare type of glial tumor, histologically classified into two types with different prognoses: high-grade and low-grade. Using both CT and MRI, Kurokawa and coworkers described detailed neuroimaging characteristics, including tumor location, margin status, morphology, CT attenuation, MRI signal intensity and tumor enhancement pattern [90]. 

##### Metastases

As mentioned above, adopting a standardized workflow is essential to improve the quality and generalizability of the radiomic models. Validations are, therefore, necessary to further increase both the sensitivity and specificity of the procedures. 

Li and colleagues evaluated the possible contribution of radiomics in differentiating HGG from brain metastases as well as the methodological quality of radiomics studies [91]. They reported that the overall sensitivity and specificity of radiomics for differentiating HGG from brain metastases were high (84%).

Qin and collaborators studied the efficiency of sensitivity-weighted imaging (SWI) in the differential diagnosis of HGG recurrence from radionecrosis and brain metastases [37] (Table 1). They reported that semiquantitative analysis by SWI was feasible for the differential diagnosis between recurrence and radionecrosis, but not for identifying metastases. Any diagnostic suggestions of brain metastasis carried out via SWI MRI after RT treatment should, therefore, be taken with caution.

Voicu and colleagues studied the diagnostic efficacy of MRI diagnostic algorithms in distinguishing between HGG and brain metastases [38] (Figure 4E–H) (Table 1). In this investigation, a qualitative analytical algorithm achieved similar results compared to the semi-quantitative approach. However, the use of quantitative, data-driven algorithms showed the best diagnostic ability. 

Hydrogen proton MRS (^1^H-MRS) imaging can noninvasively visualize tissue biochemical information in vivo and has been applied to identify and diagnose intracranial tumors. Wang and colleagues evaluated the ability of ^1^H-MRS in the identification and diagnosis of intracranial tumors via meta-analysis. They concluded that ^1^H-MRS could provide metabolic information on different intracranial tumors and effectively diagnose and differentiate glioma and metastasis [92].

Farche and colleagues reported a sensitivity of 73.3% and a specificity of 74.2% in distinguishing brain metastases from HGGs using ^1^H MRS [39] (Table 1).

##### Pseudoprogression and other Post-Treatment Effects

Pseudoprogression (PsP) or radiation necrosis (RN) can occur frequently after cranial RT and is sometimes difficult to distinguish from progressive disease (PD). Bodensohn et al. studied the diagnostic accuracy of MRI-based contrast clearance analysis (CCA) in this clinical context [40] (Figure 4I–L) (Table 1). CCA has proven to be a useful method to distinguish PsP or RN from PD after cranial RT, especially in patients with secondary tumors after radiosurgical treatment.

Mansour and coworkers described one case of pseudoprogression after intraoperative radiotherapy (ioRT) and regorafenib therapy in a patient with recurrent anaplastic astrocytoma [41] (Table 1). Relatively recently introduced therapies such as regorafenib may impact HGG MRI, as the MRI pattern in HGG imaging is changing and a multimodal approach becomes important. In particular, when using anti-vascular endothelial growth factor (VEGF) drugs, SWI may have a crucial role in identifying therapy-related hemorrhagic phenomena. 

In a cohort of patients diagnosed with HGG, radiomics and machine learning methodologies were reported as able to help predict the development of pseudoprogression from pre-treatment MR images, thus potentially allowing to reduce the use of biopsy and invasive histopathology [42] (Table 1).

The utility of DCE-Enhanced MR perfusion to distinguish between treatment damage and tumor recurrence in HGG was studied by Dündar and colleagues [43] (Table 1). To differentiate PT from TM, the authors suggested that the tissue permeability and microcirculation parameters Ktrans (volume transfer constant from the plasma compartment to the extravascular extracellular space), Kep (rate constant for transfer from extravascular extracellular space to the blood compartment), IAUC (initial area under the enhancement curve) and subtracted values of Ktrans and IAUC from nearby non-enhancing area (NNA) and contralateral normal appearing, non-enhancing area (CLNA) may be used. 

Flies et al. also studied the diagnostic value of conventional MRI features to differentiate disease progression from treatment-induced damage in HGG [44] (Figure 5A–D) (Table 1). This study proved that, in HGG patients undergoing RT, radiation dose, longer progression time, marginal enhancement, and isointense apparent diffusion coefficient (ADC) signal can distinguish disease progression from the damage induced by the treatment.

The concordance between different diagnosticians using visual qualitative assessments of MRI images, compared to the concordance of different quantitative perfusion measurements, was studied by Zakhari and colleagues (Figure 5E–G) (Table 1). While quantitative measurements of DSC and dynamic CE (DCE) perfusion maps showed satisfactory inter-rater agreements, qualitative assessment using only conventional MRI images had an inferior inter-observer agreement and was insufficient for a correct diagnostic interpretation [45].

The use of diffusion tensor imaging (DTI) to study longitudinal changes in the normal brain tissue of GB patients undergoing modern RT with volumetric modulated arc therapy (VMAT) or helical tomotherapy was investigated by Rydelius and colleagues [46] (Figure 5H–K) (Table 1). These authors observed longitudinal changes in fractional anisotropy (FA), mean diffusivity (MD) and radial diffusivity (RD), but only in a limited number of brain structures, and the changes were smaller than expected from the literature. Current modern RT techniques may cause less damage to normal tissue than in the past. 

#### 3.1.4. Specific Aspects

##### Pediatric Studies

Diffuse intrinsic pontine glioma (DIPG) remains a clinical–radiological diagnosis with little possibility of histological diagnosis. It is, therefore, important, especially for treatment purposes, to make a reliable distinction between DIPG and other pontine tumors with potentially more favorable prognoses. Kim and colleagues discussed the clinical, radiological, and treatment-related factors that may influence survival in new DIPG patients treated with RT (Figure 5L,M) (Table 1). Unfortunately, there are no significant improvements in the OS of patients with this pediatric brain tumor [47]. However, although DIPG patients have a median survival of thirteen months, there are also long-surviving patients who often have a higher age at diagnosis and ATRX mutation rates (Table 1) [48]. The need for a centralized review of DIPG cases has been emphasized for a correct and homogeneous diagnosis [49] (Table 1).

Diffuse midline gliomas (DMG) are often characterized by H3 K27 mutations and often have increased malignancy in these cases. A biopsy is not always possible to assess the H3 K27M mutation status, and attempts to identify the H3 status via MRI radiomics techniques are ongoing [50] (Table 1; section H3 K27M mutation identification).

Rodriguez and collaborators summarized the results of the HERBY study which provided these pediatric tumors with a large amount of radiological, histological–genotypic and survival data [51] (Table 1). The thalamic involvement of DMG ranged from localized partial thalamic to holo- or bi-thalamic with diffuse contiguous spread, and the degree of malignancy was little affected by the presence of H3 K27 mutations. In contrast, leptomeningeal dissemination and the surgical resection of less than 50% of the tumor were adverse factors in survival.

Bathia and colleagues compared total sodium concentrations (TSC) between pediatric glioma and non-neoplastic brain tissue using quantitative ^23^Na MRI. They were also able to improve the visualization of tissue-bound sodium concentrations (BSC) via double echo ^23^Na MRI [52] (Figure 6A–C) (Table 1). Higher sodium concentration values were found in pediatric gliomas than in non-neoplastic tissues. 

As aforementioned, distinguishing pseudoprogression following RT from actual tumor progression via imaging is often difficult, and this problem can be the cause of potentially useless or even harmful treatments. In a study of pediatric LGG, radiological differentiation criteria were developed (Table 1). Pseudoprogression was found to be frequent and independent of the RT technique used [interstitial iodine-125 RT (IS), photon beam RT (XRT), or proton beam RT (PBT)] [53].

Pediatric patients with optic pathway gliomas (OPG) often undergo multiple and frequent follow-up MRI examinations with gadolinium contrast. In several patients, the tissue accumulation of gadolinium has been found and the possible benefit/toxicity relationship must be carefully evaluated. To this end, Maloney and collaborators investigated, with a double-blind non-inferiority study and multiple examiners, if the contrast with gadolinium is necessary in OPG radiology (Table 1). A non-gadolinium-based contrast agent-enhanced protocol was found to be non-inferior to a gadolinium-based contrast agent-enhanced protocol for the follow-up of pediatric OPGs [54]. 

##### Surgical Planning

Accurate modelling of the relationship between the areas of tumor infiltration and the involvement of the eloquent areas of the connecting fibres is crucial to guarantee the maximum effectiveness and minimal collateral damage of surgical resection. Neurosurgeons usually detect an eloquent area using functional MRI and identify a connecting fibre using DTI. Although, during surgery, the accuracy of neuronavigation may be decreased due to the displacement of parts of the brain, information about their exact location can be updated by intraoperative MRI, thus facilitating subsequent surgical steps. Various intraoperative modalities can also be used to guide the surgeon’s hand such as neurophysiological monitoring (e.g., awake surgery, motor and sensory evoked potentials) and photodynamic diagnosis, which can identify areas with HGG cells. Matsumae and colleagues have discussed such intraoperative diagnostic techniques, along with their reliability and safety [93]. 

Conventional MRI may not adequately visualize areas of HGG tumor infiltration that do not take contrast. Diffusive kurtosis imaging (DKI)-derived parameters were used in a group of patients with preoperative HGG to assess proliferative activity in the boundary zone between solid tumor and peritumoral edema for treatment-planning purposes and follow-up [55] (Table 1). 

##### IDH Mutation Identification

Hagiwara and collaborators used the multiparametric MRI voxel-wise clustering method to determine the status of the IDH1 gene in HGG [56] (Figure 6D,E) (Table 1). This machine learning methodology made it possible to distinguish the mutational state of the IDH gene and the related metabolic state of tumors with good efficiency. For the same purpose, Wang and collaborators used a radiomics model based on DCE-MRI and diffusion-weighted imaging (DWI) in estimating IDH1 mutation and angiogenesis [57] (Table 1). Similarly, Cindil et al. evaluated the diagnostic performance of DWI and DSC-MRI parameters in the noninvasive prediction of the IDH gene status in HGGs [58] (Table 1). The rCBV and per cent signal recovery (PSR) parameters obtained by DSC-MRI were useful for predicting the IDH mutation status in HGGs. Thorough standardization of the method, through extensive clinical use, was indispensable.

Petridis and colleagues studied the use of resting-state blood oxygen level-dependent (BOLD) functional MRI (fMRI) to detect vascular asynchrony between HGG IDH-mutated tumors and the surrounding normal brain tissue (Table 1). Asynchrony in vascular dynamics, measured by resting-state BOLD fMRI, correlated with tumor volume and provided the radiographic delineation of HGG IDH mutated tumor boundaries [59]. Hou et al. also demonstrated that resting-state fMRI can be used to differentiate between HGG and LGG [94]. Hypoxia and tissue acidosis are often found in HGG. Yao and collaborators used an MRI technique sensitive to pH and oxygen tension to characterize some genotypic states including IDH mutation, 1p/19q co-deletion and epidermal growth factor receptor (EGFR) amplification, also studying their prognostic value [60] (Figure 7A–D) (Table 1).

Kikuchi et al. investigated whether quantitative relaxometry under synthetic MRI (syMRI) could differentiate IDH mutant HGG. Greater sensitivity was observed in comparison to qualitative T2 fluid-attenuated inversion recovery (FLAIR) [61] (Figure 7E–I) (Table 1). Quantitative relaxometry using syMRI was also proposed to differentiate astrocytomas from oligodendrogliomas. 

Li and collaborators proposed that the number of blood vessels calculable via the 3D brain volume CE (3D-BRAVO) sequence could be useful in differentiating the IDH1 status of HGGs [62].

##### H3 K27M Mutation Identification

As aforementioned, H3 K27M mutant DMG show an aggressive course. Kathrani and collaborators reported that DWI and PWI MRI characteristics of DMG may help to determine H3 K27M mutational status in DMGs preoperatively [63] (Table 1).

Guo and collaborators analyzed different radiomic models across MRI sequences and machine learning techniques to predict H3 K27M mutant status in DMG (Figure 7J) (Table 1). Multiparametric radiomic models could help predict H3 K27M mutant status but the predictive value was highly variable based on the sequences and machine learning models used [64]. 

### 3.2. PET 

#### 3.2.1. Preclinical Studies

PET imaging is increasingly used to obtain information on the metabolic status of HGG, which can valuably contribute to the diagnostic work-up. The most used radio-metabolic tracers are ^18^F-FDG and radiolabeled amino acids [11C-methionine (MET), 18F- dihydroxyphenylalanine (DOPA), ^18^F-fluoro ethyl tyrosine (FET)] and ^68^Ga-GaDOTA-somatostatin receptor (SSTR)] targeting glucose, L-amino acid metabolism and the expression of somatostatin receptors, respectively. 

The preclinical use of PET imaging using the radiotracers ^18^F-DPA-714 and ^18^F-FDopa was investigated by Clément et al. for the non-invasive detection of the IDH1 mutation in HGG models [65] (Figure 8A) (Table 1).

Ranjbar et al. have studied, in animal HGG models, the expression of the HER2 receptor, which plays an important role in the tumorigenesis and tumor progression of a wide range of tumors, using the radiotracer ^68^Ga-GaDOTA-SSTR)] [66] (Table 1).

#### 3.2.2. Clinical Studies

Since multiparametric MRI and ^18^F-FET PET are complementary imaging techniques, the information obtainable with both techniques is additive in diagnosing HGGs. For example, in distinguishing between HGG and PCNSL, the quantitative analysis of the tumor uptake of ^18^F-FDG and of the peritumoral non-uptake region combined with conventional MRI analysis has proven to be superior to the single imaging technique [67] (Table 1). Hybrid PET/MRI may, therefore, enable the simultaneous and non-invasive assessment of morphological, functional and metabolic parameters within the brain [95]. Further, the use of radiolabeled amino acids such as ^11^C-MET and ^18^F-FET can give indications not only on areas of increased proliferative metabolism but also on areas of pseudoprogression, which often represent a diagnostic problem in the follow-up of treated patients [96,97,98,99]. One further radiotracer for HGG tumors is 3,4-dihydroxy-6-^18^F-fluoro-L-phenylalanine (^18^F-DOPA), which crosses the blood–brain barrier and shows high uptake in neoplastic tissues but low uptake in normal tissues. PET analysis with ^18^F-FDOPA and MRI can non-invasively indicate the presence of IDH mutations and 1p/19q codeletions [69] (Table 1). Breen and colleagues confirmed that the PET- and MRI-guided re-irradiation of recurrent HGG with 18F-DOPA was safe and did not harm patients significantly [70] (Table 1).

^18^F-fluoroboronotyrosine (FBY) is a neutral amino acid transporter 1 (LAT-1)-dependent boron-derived tyrosine with diagnostic and therapeutic potential. The imaging characteristics of FBY PET in HGGs were studied by Kong and colleagues [71] (Table 1). High radiometabolic activity of this tracer was observed in several primary and recurrent HGGs. The tumor/fund ratio facilitated the evaluation of malignancy and possibly planned boron neutron capture therapy.

Kumar et al. described the expression of the prostate membrane antigen PSMA in recurrent HGG using Glu-NH-CO-NH-Lys-(Ahx)- [^68^Ga (HBED-CC)]-PET (^68^Ga PSMA). The tumor/fund ratio facilitated the evaluation of malignancy and possibly planned boron neutron capture therapy [72] (Table 1). 

7T MRS (see above) was compared with PET using the radiolabeled amino acids glutamine and glycine in a cohort of glioma patients [73] (Figure 8B) (Table 1). Good diagnostic concordance between the two techniques was observed, higher than what had been possible in the past with lower magnetic fields. 

Effectively determining disease extent is particularly important in diffuse and multiple gliomas. PET using the radiotracer ^18^F-fluorocholine has been suggested to be a valid imaging technique in HGG. ^18^F-fluorocholine PET has been rated superior to MRI in assessing the most likely site of recurrence [100]. Good diagnostic concordance between the two techniques was observed, higher than what had been possible in the past with lower magnetic fields.

### 3.3. Other Modelling Advances

#### 3.3.1. Preclinical Studies

The maintenance of telomeres by the reactivation of telomerase reverse transcriptase (TERT) expression is a characteristic of many tumors. Non-invasive imaging determination of TERT reactivation can indicate the tumour’s malignancy and resistance to therapy. Batsios and colleagues studied the association between TERT and metabolism in patient-derived preclinical tumor models (Table 1) [74]. 

The expression of TERT in HGGs correlated with an elevated level of NADH and the level of [U-^2^H]-pyruvate correlated with the early response to therapy, before the onset of chromosomal alterations, thus providing a biomarker of early response to therapy.

#### 3.3.2. Clinical Studies

As aforementioned, the genomic profiling of HGGs may contribute significantly to determining the degree of malignancy. The histopathological features characterizing the previous classification of WHO grade IV consisted of microvascular proliferation and necrosis development. The HGG classification published in 2021 no longer considers these histopathological features as essential to classify a grade IV glioblastoma (GB) [75] (Figure 8C,D) (Table 1). The IV level of malignancy can now be assessed even in the absence of necrosis and neoangiogenesis, except for the presence of some molecular markers such as wild-type IDH gene associated with the mutation of the TERT promoter or the amplification of the EGFR or both chromosome 7 amplification and chromosome 10 deletion (molecular GB). Despite multiple histological features similar to lower-grade gliomas, patients with molecular GB present clinical courses similar to classic GB. It has been hypothesized that these tumors are early histological GBs that will develop classic histological features over time [76] (Table 1). Indeed, our knowledge regarding the details of the exact clinical, radiographic, and histopathological findings associated with these tumors remains limited [77] (Table 1). This situation would suggest the need for biopsy and the subsequent analysis of the molecular features associated with any astrocytoma, regardless of presenting characteristics. Molecular profiling would also be important to guide the enrollment of HGG patients in early-phase clinical trials (Table 1). 

Distinguishing between HGG and brain metastasis from an extra-axial primary tumor is often difficult radiologically due to imaging similarities. A possible difference is in the peritumoral area which, in the case of HGG, is often infiltrated by neoplastic cells. In contrast, in the case of metastasis, it is only affected by pure oedema, without cellular infiltration. Marginean and collaborators proposed that these differences in the peritumoral zones could be exploited through computed tomography texture analysis to radiologically distinguish the two types of lesions (Table 1) [79] (Figure 9A) (Table 1). The texture characteristics may show a more heterogeneous content in the patient with HGG, probably due to the local infiltration of neoplastic cells, compared to the patient with brain metastasis.

Numerous studies have confirmed the usefulness of 5-ALA in identifying tumor tissues during surgical resection. However, there is no extensive information on this usefulness in the case of the resection of an HGG recurrence, which is often contaminated by necrotic tissues or pseudoprogression following the radio and chemotherapy treatments to which the primary tumor has been subjected. Ricciardi and collaborators discussed these aspects after performing a meta-analysis of the relevant published studies [101]. 5-ALA allows better fluorescence visualization in HGG but not in LGG. The fluorescence induced by 5-ALA appears to depend on the heme biosynthesis pathway, which allows the transformation of 5-ALA into fluorescent protoporphyrin IX, but the details of this mechanism are unknown. Mischkulnig et al. compared the mRNA and protein expression of intramitochondrial heme biosynthesis enzymes/transporters in glioma tissue samples showing different fluorescence levels in the presence of 5-ALA [80] (Figure 9B,C) (Table 1). The heme biosynthetic pathway activity was greater in fluorescent gliomas with the upregulation of PpIX-generating enzymes and decreased PpIX efflux mediated by the ABCG2 membrane pump. Those factors of intramitochondrial heme biosynthesis can be targeted for intervention to enable improved 5-ALA visualization even in non-fluorescent tumors such as LGG.

Microscopic vascular events, such as neovascularization and neurovascular uncoupling, are common in patients with HGG, and mapping the remodeling of this cerebrovascular network may aid in assessing cerebrovascular damage in HGG. Cai et al. studied the aspects relevant to tumor aggressiveness using an analysis of the blood oxygenation level at rest [81] (Table 1). HGG induced grade-specific cerebrovascular dysregulation in the entire brain.

Gupta et al. proposed the utility of systemic inflammatory biomarkers in distinguishing HGG from PCNSL (Table 1). 

Molecular GB (i.e., without histological examination but with the molecular characteristics of IDH-wild-type GB) often takes little contrast and its correct diagnosis can be mistaken. Mesny and colleagues evaluated the diagnostic value of gyriform infiltration as an imaging marker for molecular GB [83] (Table 1). Mesny and colleagues proposed that gyri infiltration may represent an imaging marker capable of contributing to the correct diagnosis of molecular GB.

Kraus et al. described a patient presenting with two histopathologically distinct gliomas (one oligodendroglioma and one GB) [102]. Genomic diversity was highlighted by DNA–methylation profiling, indicating that this type of analysis can contribute to the correct diagnostic classification of HGG, in addition to the response to treatment with TMZ.

HGGs account for approximately 15% of all pediatric brain tumors, and approximately 10–12% of them occur in children under the age of five at the time of diagnosis. The latter, which can show prolonged survival, are characterized by great heterogeneity of lesions with different molecular profiles, which can explain, at least in part, the different survival outcomes observed. Di Ruscio and collaborators reported their institution’s experience between 2011 and 2021, during which 11 children under five years of age with newly diagnosed HGG were followed [84] (Figure 9D–L) (Table 1). Molecular investigations played a fundamental role in the diagnostic process and the therapeutic decision for those patients. 

Tumefactive demyelinating lesions (TDLs) share clinical and imaging characteristics with HGGs. French et al. proposed a diagnostic path that may allow us to distinguish the two types of lesions and choose the appropriate type of treatment [85] (Figure 9M–R) (Table 1). Suppose MRI shows other lesions typical of MS and specific oligoclonal bands (OCBs) are found in the cerebrospinal fluid (CSF) via lumbar puncture (LP). In that case, patients should undergo a short course of steroids to seek improvement clinically. Patients who continue to worsen, who show no other lesions on MRI, or in whom the LP is negative for CSF-specific OCB, should be considered for biopsy with histology. This pathway can provide patients with the best diagnostic and neurological preservation possibilities.

## 4. Models for the Prognosis of HGG

### 4.1. Preclinical Studies

Planeta and colleagues applied X-ray fluorescence microscopy based on synchrotron radiation to study the micro changes of the rat brain in which HGG develops [103] (Figure 10A) (Table 2). The changes in the animal brain implanted with two different stabilized lines of human HGG and with an HGG (GB) taken directly from a patient were compared. In each case, the extent and intensity of the micro changes in the brain parenchyma were strongly linked to tumor development. Tumors developed in rat brains were characterized by the accumulation of Fe and Se, while peritumoral tissue by the accumulation of Cu.

**Table 2 cancers-16-01566-t002:** Models for the prognosis of HGG.

Major Finding	Experimental System	Ref.
Increased accumulation of Fe and Se in tumor and Cu in peritumoral tissue in rodent models.	Orthotopic rat models of GB.	[103]
LAT1 as a new marker for GICs.	LAT1+ and LAT1- glioma cells sorted by flow cytometry.	[104]
Circulating miR-181a/b, miR-410 and miR-155 as diagnostic and prognostic biomarkers in HGG.	Determination of pre- and postoperative plasma levels of miR-181a/b, miR-410 and miR-155 in 114 HGG patients, 77 LGG patients and 85 healthy volunteers as control group.	[105]
Multi-parameter MRI as a non-invasive method for the prognosis of DMG.	84 patients with DMG including 40 patients with OS > 12 months and 44 patients with OS < 12 months.	[106]
The S100 protein signature in the HGG patients’ prognosis.	Determination of the expression profiles of 17 *S100* family genes in glioma.	[107]
The relationship between glioma angiogenesis and the malignant phenotype, immune characteristics, and prognosis.	An angiogenesis pathway score assessing the status of intra-glioma angiogenesis using public datasets.	[108]
Dismal prognosis in *H3.3 G34*-mutant glioma patients.	30 adults with *H3.3 G34*-mutant diffuse gliomas.	[109]
rs-fMRI may identify neural correlates for cognitive and daily functioning in glioma patients.	22 patients with diffuse gliomas who completed treatment within the past 10 years.	[110]
Assessment of side effects of RT should include depression.	15 patients with HGG receiving standard radio(chemo)therapy.	[111]
The association between peripheral blood tests, cMRI and prognosis.	131 GB patients.	[112]
Basal ganglia iron levels as a biomarker in glioma prognosis and treatment.	59 patients with brain lesions.	[113]
A “DeepRisk” learning model predicting glioma survival from whole-brain MRI.	1556 patients with diffuse gliomas.	[114]
A nomogram based on MRI radiomics and clinical features for predicting *H3 K27M* mutation in pediatric HGGs.	107 patients with pHGGs with a midline location of the brain including 79 patients with *H3 K27M* mutation.	[115]
A nomogram based on clinical pathology, genetic factors, and MRI predicting early recurrence of HGG.	154 patients with HGG classified into recurrence and nonrecurrence groups based on the pathological diagnosis and RANO criteria.	[116]
A novel ARG-related risk signature as a prognostic marker.	1738 glioma patients collected from three public databases.	[117]
A 14 radiomic features-based prognostic model constructed from preoperative T2-weighted MRI images.	652 glioma patients across three independent cohorts.	[118]
A combination of ^18^F-DOPA PET and MRI for distinguishing TP from TIE after RT.	76 patients showing at least one gadolinium-enhanced lesion on the T1-w MRI sequence.	[119]
The relationship between CE in MRI and fluorescence during surgery in glioma patients.	179 patients with newly diagnosedgrade II and grade III gliomas who received 5-ALA for resection.	[120]
Differences in survival between patients with primary and secondary GS.	94 GS patients; 70 with primary disease and 24 with secondary.	[121]
The performance status of elderly patients is the most important prognostic factor.	198 patients with grade IV glioma over 65 years at the time of diagnosis; grade III gliomas with nonmutated *R132HIDH1* and radiographically only diagnosed gliomas.	[122]
The association between the Ki-67 index and edema.	MRI studies of 70 patients with GB acquired up to one week before surgery.	[123]
Elevated prognostic capacity of imaging-based risk stratification in patients with diffuse glioma, NOS.	220 patients classified as diffuse glioma, NOS.	[124]

**Figure 10 cancers-16-01566-f010:**
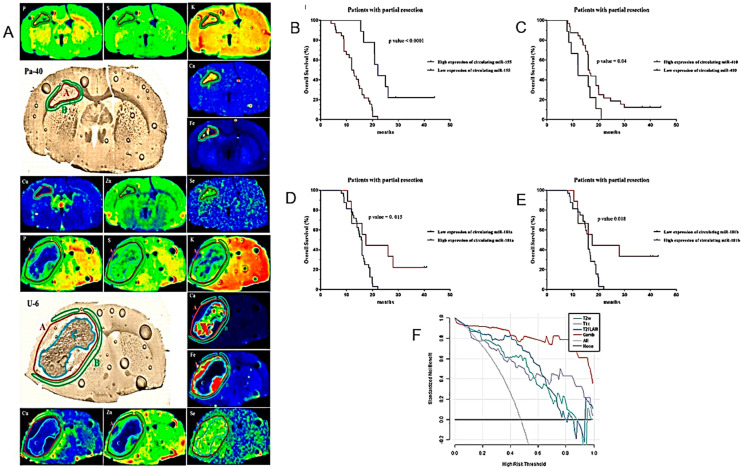
Models for the prognosis of HGG I. (**A**). Elemental anomalies accompanying HGG development within an animal’s brain might facilitate our understanding of the pathogenesis and progress of the tumors and also determine the potential biomarkers of their extension. Regions of tumor (red A), surrounding tissue (green B) and tumor debris (blue C) marked in the maps of elemental distribution as well as microscopic images of brain slices taken from selected animals implanted with patient (Pa-40)-derived GB cells (**top**) and U87 (U-6) established GB cell line (**bottom**) groups. (**B**–**E**). Kaplan–Meier curves of OS for HGG patients based on the circulating miR-155, miR-410, miR-181a and miR-181b signature in patients with partial tumor resection. (**F**). Decision-curve analysis of four radiomic models for the non-invasive prognosis of the OS of patients with midline gliomas. The *x*-axis represents the threshold probability, and the *y*-axis represents the net benefit. The combined model (red line) had a higher net benefit and better application value for predicting the survival time of patients with midline gliomas (after [103,105,106] with permission).

Glioma-initiating cells (GICs) are considered to be the origin of HGG development, both primary and recurrent. Wang and colleagues suggested that the essential amino acid transporter LAT1 (CD98) might be a potential biomarker of GICs [104] (Table 2). LAT1 had previously been linked to the invasive capacity of HGGs. Wang and colleagues found that LAT1-expressing cells possess stemness characteristics and the ability to promote HGG development, suggesting LAT1 as a possible additional biomarker of GICs. The microenvironment of the central nervous system (CNS) is generally considered not permissive for the development and invasiveness of HGG, although this control system can, unfortunately, fail. De Luca and colleagues discussed several aspects of the anatomical and biological basis of gliomagenesis, including neural stem cells, the spatiotemporal diversity of astrocytes and microglia, and the extracellular matrix and peritumoral environment [125]. The analysis suggested that GICs should not be considered the only source of cells capable of initiating HGG development and recurrence. 

The downregulation of miR-181a/b and miR-410 and upregulation of miR-155 have been shown to play a role in the development and progression of HGGs. Wu, J. et al. proposed that circulating miR-181a/b, miR-410 and miR-155 could be used as non-invasive diagnostic and prognostic biomarkers in HGG [105] (Figure 10B–E) (Table 2).

Using the multiparametric MRI features of HGGs, Deng and co-workers developed four radiomic models for the survival prognosis of midline glioma patients [106] (Figure 10F) (Table 2). All four models taken individually were adequate for sensitivity and specificity. A model combining all four features performed best and offered the highest application value for the noninvasive prognosis of patients with midline gliomas.

The S100 protein family comprises 25 calcium (Ca^2+^)-binding proteins with high structural and sequence similarity. The deregulation of this family of proteins can alter various proliferation control mechanisms, increasing the possibility of tumor development and progression. For example, the family member S100A4, which is secreted by tumor and stromal cells, stimulates tumor angiogenesis, acting synergistically with VEGF. Studies with animal models obtained through human tumor xenografts have suggested that the in vivo expression of the S100A4 member can favor the development and invasiveness of various types of tumors, including melanoma and pancreatic carcinoma. Using a dataset obtained from the Chinese glioma genome atlas (CGGA) and the cancer genome atlas (TCGA), Wang, LJ et al. proposed a prognostic model of HGGs based on the expression profiles of seventeen S100 family genes (including S100A4) in relation to the tumor microenvironment [107] (Figure 11A–G) (Table 2).

Correct surgical/RT planning is based on defined anatomical considerations on tumor development and progression. Zhang, Q. et al. developed an angiogenesis score using gene set variation analysis (GSAV) to assess the status of intra-HGG angiogenesis in TCGA, CGGA mRNAseq_325, CGGA mRNA-array and NCBI’s gene expression omnibus (GEO) GSE16011 datasets Correct surgical/RT planning is based on defined anatomical considerations on tumor development and progression. Zhang, Q. et al. developed an angiogenesis score using gene set variation analysis (GSAV) to assess the status of intra-HGG angiogenesis in TCGA, CGGA mRNAseq_325, CGGA mRNA-array and NCBI’s gene expression omnibus (GEO) GSE16011 datasets [108] (Figure 11H–O) (Table 2). The scores thus calculated in the angiogenic development of the tumor can help define the invasiveness of the tumor, its sensitivity to chemo- and RT and, ultimately, the prognosis of the patients.

### 4.2. Clinical Studies 

Similar to the above-discussed contributions to image-based modelling for the diagnosis of HGG, advances in AI may offer progress in detailing the structure and pathology of those tumors, previously beyond the scope of human experience alone; ultimately this contributes to improving patient prognosis. These possibilities pass through a machine learning process involving the steps of image preprocessing, tumor segmentation, feature extraction and the construction of classifiers which may finally lead to the development of prognostic models [126]. Wang et al. described the characteristics of adult HGG with the H3.3 G34 mutation (Table 2). Thirty adults with diffuse gliomas with the H3.3 G34 mutation were retrospectively examined clinically and pathologically. A higher frequency of the loss of Olig-2 expression, TP53 mutation, ATRX mutation, PDGFRA mutation, and MGMT promoter methylation was found in these HGGs compared to patients with H3.3 WT gliomas. However, no TERT promoter mutation and only one case of EGFR amplification were detected, the frequencies of which were significantly higher in the wild-type IDH/H3 cohort. Patients with the H3.3 G34 mutation had a worse prognosis than WT patients.

It has been shown that the expression levels of the CUL2 gene, which encodes the cullin2 protein in the cullin2-RING E3 (CRL2) DNA ligase complex, can predict the radiosensitivity and prognosis of HGG patients. The expression levels of CUL2 are tightly regulated with its copy number variations (CNVs). Zander and colleagues developed artificial neural networks (ANNs) for the noninvasive pretreatment evaluation of HGG patients integrating clinical measurements, genetic data, and image data [127]. 

Resting-state functional magnetic resonance imaging (rs-fMRI) was used to study alterations in the functional connectivity (FC) associated with cognitive function in long-surviving HGG patients [110] (Figure 12A–E) (Table 2). Some rs-fMRI parameters potentially related to cognitive and daily functioning in HGG patients were described.

Schiavolin and collaborators proposed a possible standardization of outcome parameters to compare studies on cognitive impairment in three categories of patients: affected by HGG, meningioma or subject to cerebrovascular surgery. Identifying common outcome measures was the first step in this standardization process [128]. RT in patients with brain tumors may affect the structure of the hippocampus and cause dyscognitive side effects that may contribute to the depressive symptoms often present in these patients. Prompt and effective pharmacological intervention for these symptoms can improve the quality of life of HGG patients. Donix et al. highlighted that the evaluation of the side effects of RT concerning memory should include any depressive symptoms [111]. Preoperative conventional peripheral blood and MRI tests can help inform the prognosis of patients with HGG. Rao and colleagues proposed calculating a risk score based on the results of a preoperative peripheral blood test and conventional MRI, then developing a prognostic nomogram for HGG (Figure 12F–K) (Table 2). VEGFA was the most important gene related to peripheral blood tests significantly associated with an unfavorable prognosis and the nomogram contributed to the prognostic classification of the patient [112]. 

Invasive growth along white matter (WM) tracts is an important issue in HGG surgery. Wang and collaborators discussed the imaging and histological characteristics of HGG patients with WM tract invasion and summarized the possible molecular mechanism [129]. The relationship between the molecular profile of the tumor and the radiation fields used in RT with the invasion of the white matter by the HGG tumors was also discussed. 

HGGs have been found to alter iron metabolism and transport, thus increasing the intracellular iron storage possibly supporting tumor growth. Reith and colleagues determined basal ganglia iron concentrations in HGG patients using deep neural network-trained quantitative susceptibility mapping (QSM) [113] (Figure 12L,M) (Table 2). The iron content in the basal ganglia correlated with the malignancy of the tumor and the authors suggested using the level of deposited iron as a negative prognostic biomarker.

A prognosis model for the OS of HGG patients called “Deep Risk” has been proposed, starting from MRI data on the whole brain without the need to segment it [114] (Table 2). Deep Risk achieved prognostic accuracy comparable to models built using segmented tumor images. Deep Risk had independent and incremental prognostic value over existing models and IDH mutation status. 

Wu and colleagues developed a nomogram based on MRI radiomics and clinical characteristics to preoperatively predict the H3 K27M mutation in pediatric HGGs (pHGGs) extending to the midline of the brain (Table 2) [115]. Qing and colleagues similarly developed a nomogram to predict the early recurrence of HGG using clinical pathology, genetics, and MRI data [116] (Table 2). 

Fu and collaborators have studied the role of autophagy-related genes (ARGs) in the progression of HGG. An autophagic progression risk profile was described for HGG patients, further developing a prognostic nomogram potentially useful for selecting the most effective treatment [117] (Table 2) (Figure 13A).

A preoperative T2-weighted MRI-based radiomics model was developed by Li and collaborators as a survival prognostic tool for HGG patients [118] (Figure 13B–D) (Table 2). The model was based on 14 radiomic parameters, among which the extent of macrophage infiltration was the one most influencing the prognosis. 

Similar to pediatric tumors, adult diffuse midline gliomas (aDMGs) with H3 K27M alterations are associated with relatively short survival [130]. A further negative prognostic element in these tumors is the mutation of the ATRX gene, while RT has a positive effect on survival.

^18^F-DOPA PET is used in the follow-up of HGG after RT to distinguish between RT treatment-induced lesions and tumor progression. Bertaux and collaborators compared the diagnostic and prognostic efficacy of PET-DOPA with the PET/DOPA associated with MRI, finding a significant increase in resolution in favor of the latter [119] (Table 2). (Table 2). The difficulty in treating HGGs is partly linked to their heterogeneity, which allows selected populations of tumor cells to escape the effect of therapies. The tumor microenvironment and the non-tumor cells within it also contribute to the tumor’s resistance ability in various ways. Zhou and coworkers emphasized that today’s imaging techniques assisted by AI should be developed to provide more precise information on the conformation of HGG tissues, which are important for defining the prognosis and personalizing treatments [131]. 5-ALA tumor fluorescence often correlates with the degree of malignancy. Muther and collaborators have described imaging features related to CE indicating the tumor 5-ALA level of fluorescence and its degree of malignancy (Figure 13E) (Table 2) noninvasively. 

Gliosarcoma (GS) is a subtype of GB with sarcomatous features. GS tend to metastasize, unlike other gliomas, with lower 5-year survival rates than GB patients. Differences in survival between patients with primary and secondary GS have been studied and described by Amer and colleagues [121] (Table 2). 

Leibetseder and collaborators studied the clinical and radiological parameters possibly influencing the prognosis of patients with brainstem gliomas (BSG). The ECOG Performance Status scale, the body mass index, the WHO grade and the ADC were proposed as possible parameters associated with the OS of BSG patients [132]. The prognosis of HGG patients is worse with advancing age. In an epidemiological study carried out on the Finnish population, Pirkallainen and collaborators confirmed that the performance status, rather than the chronological age, was one of the most important parameters influencing the choice of treatment and, ultimately, the OS of patients [122] (Figure 13F) (Table 2).

Caramanti and collaborators found a correlation between the volume of tumor-induced oedema measured by MRI and the Ki-67 proliferation index. According to this study, the two parameters might be considered in relation to each other and be prognostic of OS [123] (Figure 13G,H) (Table 2).

Some gliomas escape precise histopathological classification and are referred to as not otherwise specified (NOS). Jang and colleagues attempted to determine the prognosis of patient survival based on the imaging characteristics of these tumors. Three risk categories for NOS gliomas were identified. The predictive capacity was particularly high for long-survival prognoses (three or five years of survival) [124] (Table 2). This prognostic imaging model could also be useful to direct molecular investigations for these types of gliomas. 

## 5. Discussion 

### 5.1. Modelling the Diagnosis of HGG

The publication of subsequent versions of the classification criteria for brain tumors testifies to identifying increasingly numerous entities through the progressive advancement of molecular diagnostic techniques. Examples in adults are IDH mutant astrocytoma; IDH mutant oligodendroglioma and codeletions 1p/19q; glioblastoma, IDH-wild-type. In children, the examples are as follows: diffuse hemispheric glioma with H3 G34R mutation and HGGs with wild-type H3 and IDH genes [133,134]. Alongside molecular investigations on tissues and tumor cells, the advancement of tumor imaging techniques contributes to the identification and typing of lesions, the prediction of metastasis and, ultimately, the prognosis of patient survival. In addition to distinguishing between different morphologies of primary brain tumors, they help to distinguish tumors of different tissue origin, such as PCNSL and brain metastases, e.g., originating from lung tumors; brain lesions induced by treatments (e.g., necrosis) and pseudo-progression phenomena.

Advanced MRI techniques such as DSC, DWI, DCE, BOLD and ASL can intervene at various levels in this diagnostic process [135]. In particular, perfusion with dynamic sensitivity can contribute to the diagnostic differentiation between tumor progression and radiation damage in patients with HGG [136]. DWI combined with choline levels determined by MRS can also significantly improve the differential diagnosis of recurrent HGG and radiation damage, as well as the intraoperative characterization of tumor type and margin [137,138,139]. MR relaxometry, a quantitative imaging method that measures tissue relaxation properties, can help differentiate between gliomas and metastases and between different grades of glioma [140]. Relaxometry can help define the peritumoral areas with a greater probability of tumor infiltration and the hypoxic areas not identified by perfusion. Studies on the response to anti-tumor therapy have also suggested an association between survival and progression with tumor relaxometric profiles.

Combined with MRI, PET, especially if carried out with radiolabeled amino acids, is acquiring an increasingly relevant role in the distinction between HGG and PCNSL [141], HGG and metastases [142,143] and HGG and therapy-related damage [144,145]. 

However, the diagnosis of HGG carried out using advanced neuroradiology techniques is still in an experimental phase and lacks standard protocols. The definition of the latter is key to fully exploiting the potential of the many advanced imaging techniques currently available. Meritorious attempts in this sense are underway by the working group of the Imaging Biomarker Standardization Initiative which has proposed standardized radiomic characteristics for phenotyping based on high-throughput images [146]. The adoption of standard algorithms should significantly improve the reproducibility of AI-based radiomic models, and it represents an essential step for its widespread adoption in the clinic. Unfortunately, this is easier said than done; the definition of standardized procedures can only go through multicenter clinical studies on large samples [19,44,147,148].

### 5.2. Modelling the Prognosis of HGG 

Prognostic tests that can be carried out on liquid biopsies from which circulating cells, circulating DNA and RNA can be analyzed are under investigation, but truly informative markers for the prognosis of HGG patients are sparse [149,150]. Modified RNAs that are important factors in the metabolic cooperation between HGG and TME [151,152] have been proposed as diagnostic and prognostic markers. However, those results require large validating multicenter clinical studies [153]. While multifocal HGGs and their infratentorial recurrences (ITR) negatively influence the prognosis of HGG patients [154,155], the socioeconomic status (SES) has been indicated as a positive prognostic factor. Here again, large and well-designed studies are necessary [156]. A recent meta-analysis reported that several nutritional indicators, including the prognostic nutritional index (PNI) score and the control nutritional status (CONUT) score, as well as the assessment of sarcopenia, may contribute to the prognosis of HGG patients [157,158].

Due to the different types of evidence, indices of immunoinflammation may be prognostic markers at a more advanced stage of characterization [159]. Macrophages, nTreg and Th2 cells perform immunosuppressive functions in the TME and can be markers of tumor progression. In particular, while glioma-associated M1-type macrophages are characterized by an increased secretion of proinflammatory cytokines, such as interleukin (IL)-1ß, tumor necrosis factor (TNF), IL-27, matrix metalloproteinases (MMP), C-C motif chemokine ligand 2 (CCL2), VEGF and insulin-like growth factor 1 (IGF1), and can lead to the destruction of tumor tissue, those of the M2 type have more of an immunosuppressive action and promote tumor progression through the production of IL-10, IL-35 and transforming growth factor ß (TGF-β) [160]. The overexpression of *NFE2L2*, *NOX4* and *PD-L1* genes correlates with such immunosuppression phenotype [161]. Simple blood count indices of immunoinflammation, such as the ratio of platelets to lymphocytes, may also contribute to the prognosis of HGG patients [162]. 

### 5.3. Regulating the End of Life

Despite research efforts, the prognosis for HGG patients remains poor with a median OS of 36 and 15 months for grade III and IV tumors, respectively. In this regard, there is much to do when the end of life approaches, and this point must be discussed. A profound burden of cognitive and relational symptoms can afflict the patient’s disease course and the emotional and physical burden of their caregivers. Predominant symptoms include seizures, headaches, depression, fatigue and treatment-induced toxicity, which can be addressed through the appropriate organization of specialist palliative care services to mitigate the tremendous impacts of the disease [163]. Unfortunately, while such organization has become relatively common in systemic tumors, it remains unjustifiably limited in neuro-oncology [164]. 

Cognitive decline can heavily limit the patient’s capacity to distinguish and express preferences regarding the continuation of treatments. This is an important ethical issue involving patients, their families and healthcare professionals. It is valuable in these cases to compile, when the neurocognitive functions are still intact, a living will that details the conditions and methods chosen by the patient for the treatments to undergo at the end of life. However, although this living will has had legal value in Italy since 2017 (Law 22 December 2017, n. 219 “Provisions for informed consent and advance directives treatment”), it is unfortunately still filled out by a minority of HGG patients [165]. A much stronger communicative effort is needed on these options to make the end of life course more defined and shared between the patient, his/her family and healthcare professionals [166].

Already present in the laws of several European and overseas countries, assisted suicide choices have now also been made possible by a sentence of the Italian Constitutional Court (242/2019), which provides for it in cases where the following four conditions have been verified by a public structure of the national health service, following the opinion of the territorially competent ethics committee: 1. Intention to commit suicide, independently and freely formed. 2. Person kept alive by life-sustaining treatments. 3. Person suffering from an irreversible pathology, a source of physical or psychological suffering that he/she considers intolerable. 4. Person fully capable of making free and informed decisions https://www.cortecostituzionale.it/documenti/download/doc/recent_judgments/Sentenza_n_242_del_2019_Modugno_en.pdf (accessed on 17 April 2024). In this regard, depression is a common complication that can cause psychological severe barriers, rapidly deteriorate the patient’s quality of life (QoL) and limit the capabilities referred to in point 4. Currently, the Hospital Anxiety and Depression Scale (HADS) is the most commonly used tool to diagnose depression in HGG patients. Female gender, unmarried status, low level of education, high tumor grade, and a history of mental illness may increase the risk of depression and depressive symptoms in patients with HGG.

## 6. Conclusions

The conversion of radiological images into data (radiomics and radiogenomics) through AI may allow automatic learning processes for machines, contributing non-invasively to molecular profiling, albeit with lower resolution than histopathological techniques [167]. The most robust radio genomics acquisition at the moment is the ability to predict the mutated state of the *IDH* gene, which can significantly influence the course of HGG diseases. The low sample size of studies and the different protocols/methodologies used in clinical trials still represent formidable limits to harmonizing methodologies, comparing the results, assembling data and achieving robust, validated radiomic protocols [168]. To those aims, it is urgent to define an international consensus to which the various clinical research groups operating in the field may refer.

A number of novel prognostic markers for HGG have been proposed in the first semester of 2022 in preclinical or monocentric clinical studies. While preclinical studies provide valuable insights, the translation of findings to clinical practice may be limited due to the differences between animal models and human pathology. Randomized multicenter clinical studies validating the prognostic capacity of most markers have not yet even started and are eagerly waited for.

Research on palliative care for HGGs, and often their availability for HGG patients, is lagging. In many cases, once patients leave the hospital, they find themselves without adequate and early palliative interventions, especially when they live alone. More research is needed on the coordination of these treatments and their specific aspects. We must define the timing, context, and specific components of palliative intervention care, including effective communication protocols [169]. Bio-psycho-social (pharmacological–psychotherapeutic–sociotherapeutic) evaluation and intervention on mood disorders may improve the awareness and overall quality of life of patients with HGG and help them face the end of their life with greater serenity [170].

## Figures and Tables

**Figure 1 cancers-16-01566-f001:**
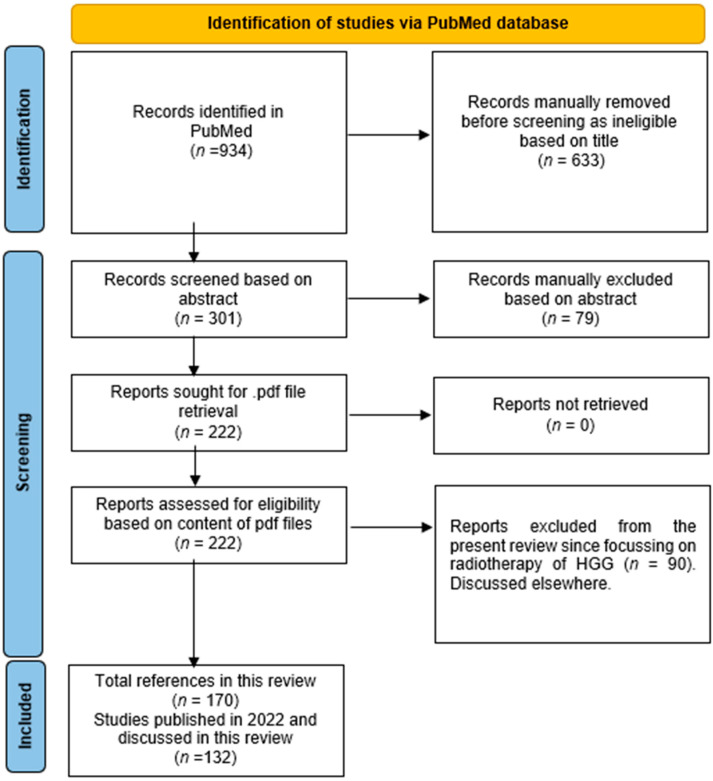
Flow diagram for reported studies (modified from [16] with permission).

**Figure 3 cancers-16-01566-f003:**
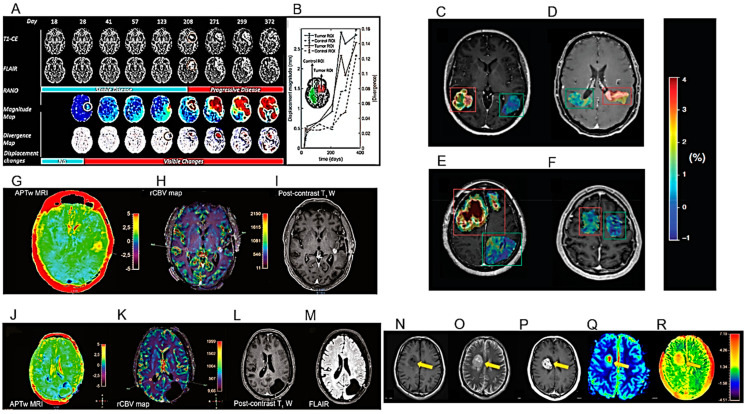
Models for the diagnosis of HGG II. (**A**,**B**). Longitudinal evolution of GB from day 18 after the start of radio–chemotherapy treatment to day 372. (**A**). According to RANO criteria, tumor progression started on day 208. However, the displacement maps show significant deformations already at day 28. Circles in orange highlight preliminary visual evidence of tumor growth. (**B**). Quantification of the displacement magnitude and absolute divergence in control and tumor ROIs, respectively. (**C**–**F**). Single transversal APTw MRI slices of four different GB patients. APTw MRI value in a color scale projected on the post-contrast T1-weighted image, within the tumor ROI only. Red and green boxes are positioned around the tumor and contralateral ROIs, respectively. Note the heterogeneous hyperintensity in the tumorous regions that show hyperintensity on the CE T1-weighted image when compared to the contralateral region (**C**–**E**). (**F**). The APTw MRI value of a non-enhancing GB. (**G**–**I**) Patient with histopathologically confirmed GB, treated with surgery and chemoradiation. (**G**). APTw MRI with hyperintense signal. (**H**). rCBV map with increased rCBV (3.34) relative to contralateral normal-appearing white matter. (**I**). Enhancement on post-contrast T1 weighted image. (**J**–**M**). Patient with recurrent GB, treated with surgery, chemoradiation and anti-angiogenic therapy upon recurrence. (**J**). APTw MRI with iso-to hypointense signal around the operation cavity. (**K**). The relative cerebral blood volume (rCBV) map does not show significantly increased rCBV (1.41) relative to contralateral normal-appearing white matter. (**L**). Post-contrast T1 weighted image demonstrates no enhancement. (**M**). FLAIR exhibits no sign of tumor progression. (**N**–**R**). A 62-year-old male patient with an HGG in the right basal ganglia. The APTw MRI signal range of the patient was larger than that on both T2WI and enhanced scans and larger than that on PWI. (**N**). T1-weighted imaging showed an iso-slightly low signal. (**O**). T2-weighted imaging showed an iso-slightly high signal. (**P**). Enhanced MR imaging showed obvious enhancement. (**Q**). PWI showed marked hyper-perfusion. (**R**). APTw MRI shows that the APT rate increased (after [31,32,33,34] with permission).

**Figure 4 cancers-16-01566-f004:**
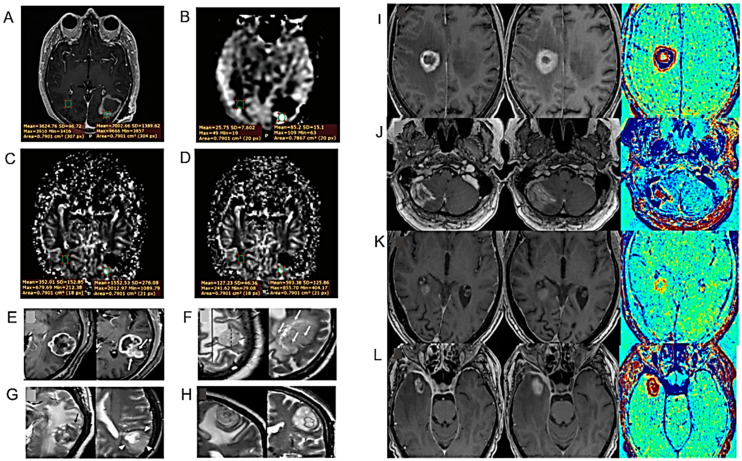
Models for the diagnosis of HGG III. (**A**–**D**). Example of ASL and DSC measurements with tumor region of interest (ROI) placement. Lesion ROI and contralateral normal-appearing white matter (NAWM) ROI. (**A**) T1W+Gd. (**B**) ASL-CBF. (**C**) DSC-rCBV. (**D**) DSC-rCBV leakage corrected. (**E**–**H**). Analytic qualitative criteria used to differentiate single metastases (MET) from high-grade gliomas (GB). (**E**). Morphology and margins characteristics on post-contrast T1-weighted (T1w) axial images of a 66-year-old man with a single brain metastasis from colon cancer (left), and in a 65-year-old man with GB (right). The metastasis has an almost spherical shape and well-defined margins. In contrast, GB exhibits an irregular shape, and areas with poorly defined margins (arrows). (**F**). Edema/lesion ratio and macroscopic vascularization on T2w axial images in a 56-year-old woman with single brain metastasis from breast cancer (left) and in a 71-year-old man with GB (right). The metastasis has a high ratio between edema (continuous caliper) and lesion (dashed caliper). No prominent vessels coursing within the lesion are seen. In contrast, GB has a relatively lower edema/lesion ratio and prominent intralesional vessels (arrows). (**G**). Lesion relationship with the cortex on T2w axial images, in a 66-year-old man with single brain metastasis (left) and in a 62-year-old man with GB (right). The metastasis shows no thickening or definite signal change of the cortex in the proximity of the lesion (arrow). However, GB causes the thickening and blurring of the cortex interface, suggesting infiltration (arrowheads). (**H**). T2-signal texture characteristics in the peri-enhancing region: coronal images of a 67-year-old woman with single brain metastasis from breast cancer (left) and a 75-year-old woman with GB (right). Light gray outlines mark the corresponding enhancing nodules as seen on post-contrast T1w images. Whereas the metastasis exhibits a uniformly bright signal in the peri-enhancing region, suggesting simple vasogenic edema, GB shows adjacent white matter signal inhomogeneity, with subtle hypo intensities (arrowheads) suggesting tumor infiltration. (**I**–**L**). Examples of contrast clearance analysis (CCA). The images show four different patients (**I**–**L**), each with a regular CE T1-MRI sequence, a late phase T1-sequence 1h after contrast media application, and their CCA (from left to right). Tumor tissue is depicted as blue in the CCA, while reactive tissue is depicted as red. (**I**). GB IDH WT: a frontoparietal lesion showing tumor tissue in a circular formation with reactive components centrally and at the lesional border. (**J**). Lung adenocarcinoma with brain metastases: a right cerebellar lesion showing tumor tissue with reactive components in the surrounding area. (**K**). GB IDH WT: a periventricular lesion showing spotted areas with reactive tissue. (**L**). Maxillary squamous cell cancer with brain infiltration: a lesion in the right temporal lobe consisting nearly entirely of reactive tissue (after [36,38,40] with permission].

**Figure 5 cancers-16-01566-f005:**
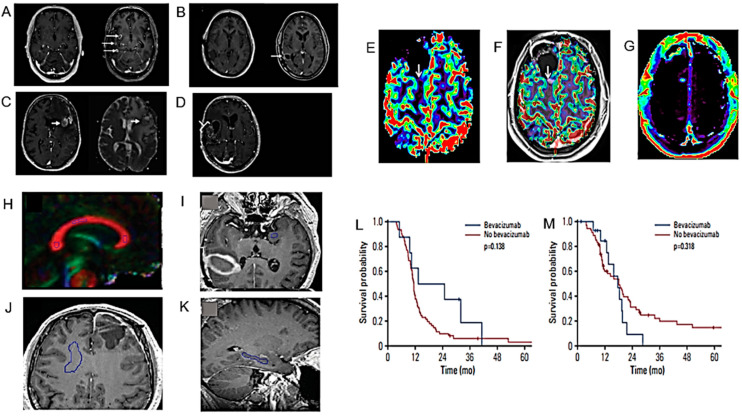
Models for the diagnosis of HGG IV. (**A**–**D**). MRI Characteristics with Significant Predictive Value. (**A**). A 64-year-old man with an isocitrate dehydrogenase (IDH)–wild-type (WT) GB treated with RT. Left: Preoperative baseline MRI. Right: Follow-up MRI (93 days after baseline) with multiple new CEs. (**B**). A 58-year-old man with an IDH-WT GB treated with TMZ-based chemoradiation. Left: Baseline MRI of the surgical cavity after resection. Right: Follow-up MRI (91 days after baseline) with increased marginal enhancement of the surgical cavity. (**C**). A 65-year-old woman with a GB, IDH status unknown, treated with RT. Left: T1-MRI with contrast agent. Right: Isointense ADC signal compared with healthy white matter. (**D**). A 66-year-old man with an IDH-mutated astrocytoma grade 4 treated with TMZ-based chemoradiation: soap bubble enhancement (small regions of necrosis). (**E**–**G**). Low corrected CBV: axial dynamic susceptibility contrast MR derived corrected CBV map (**E**) and axial post-contrast T1WI with co-registered corrected CBV map (**F**) showing low CBV in a solid enhancing nodule. CE MR perfusion derived k-trans map (**G**) showing intermediate K-trans in the same lesion. (**H**–**K**). DTI parameters in different normal-appearing structures were analyzed after defining ROIs in homogenous tissue. Three different structures of the corpus callosum were examined: splenium, corpus, and genu. (**H**). The corpus callosum on a FA-color map. The ROI placed within the structures from the left: splenium, corpus and genu, viewed at a sagittal view. (**I**). The centrum semiovale in the right hemisphere on a post-contrast T1-weighted image, viewed on the transversal plane. (**J**). The amygdala in the left hemisphere on a post-contrast T1-weighted image, viewed on the transversal plane. (**K**). The left hippocampus on a post-contrast T1-weighted image, viewed on the sagittal plane. (**L**,**M**). Bevacizumab was administered to 23 DIPG patients. Among patients who had increased post-RT necrosis, the median OS was 13.3 months and 11.4 months for those who did and did not use bevacizumab, respectively (**L**). Among patients without or with decreased post-radiotherapy necrosis, the median OS was 17.6 months and 18.1 months for patients that did and did not use bevacizumab, respectively (**M**). Relevant Kaplan–Meier plots are shown (after [44,45,46,47] with permission).

**Figure 6 cancers-16-01566-f006:**
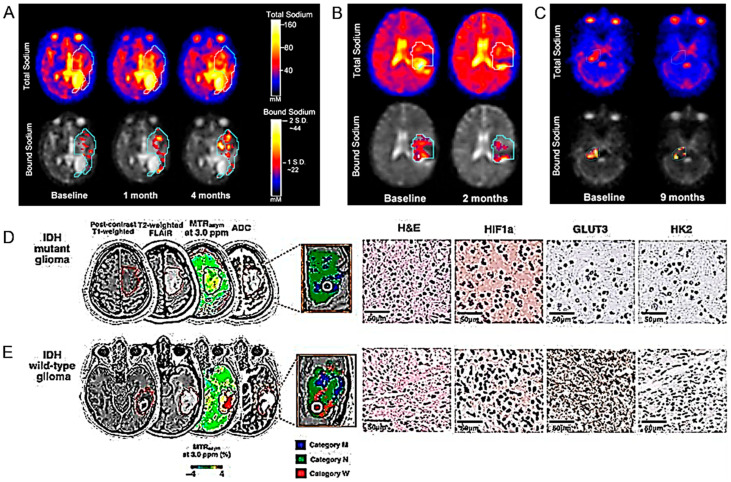
Models for the diagnosis of HGG V. (**A**–**C**). (**A**). Two-TE sodium MRI scans showing tumor progression in HGG. (**B**). Response to therapy in supratentorial astrocytoma. (**C**). Response to therapy in HGG. In the tumor regions of the bound sodium images are pixels of bound sodium concentration (vBSC) with a value greater than 1 standard deviation (S.D.) from the average vBSC value (~22 mM) over the tumor. (**D**,**E**). MR images and corresponding hematoxylin and eosin (H&E) and immunohistochemistry staining for MRI-guided biopsy targets (circles). (**D**). IDH mutant glioma for which an area with labels categorized as M, indicating the IDH mutant feature, was biopsied. Expressions of HIF1a, GLUT3, and HK2 are low in the slides from a 5 mm radius sample taken from the MRI-guided biopsy target. (**E**). IDH wild-type glioma for which an area with labels categorized as W, indicating IDH wild-type feature, was biopsied. Expressions of HIF1a, GLUT3, and HK2 are high (after [52,56] with permission).

**Figure 7 cancers-16-01566-f007:**
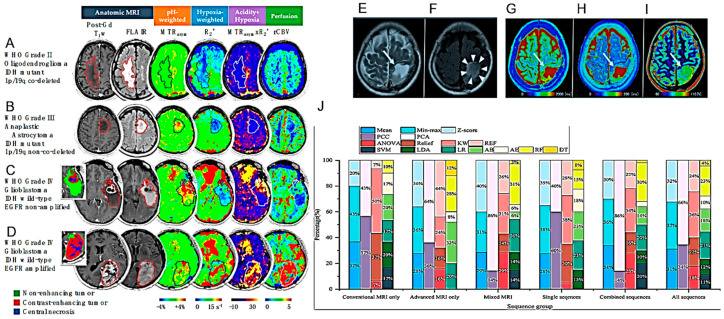
Models for the diagnosis of HGG VI. (**A**–**D**). pH- and oxygen-sensitive MR images in representative glioma patients. Four patient examples with IDH mutant 1p/19q co-deleted glioma (**A**), IDH mutant 1p/19q non-co-deleted glioma (**B**), IDH wild-type EGFR non-amplified glioma (**C**), and IDH wild-type EGFR amplified glioma (**D**) are illustrated. Tumors are outlined in each image, with segmented tumor ROIs demonstrated for patients (**C**,**D**). Regions with elevated acidity, high hypoxia, and increased perfusion in the pH-weighted images, hypoxia-sensitive images, and perfusion images are represented by red colors, corresponding to high values of MTRasym, R2′, and rCBV, respectively. Similarly, a high level of combined acidity and hypoxia is highlighted in yellow on the MTRasym × R2′ map. (**E**,**F**). Images from a 49-year-old woman with IDH-mutant diffuse astrocytoma (WHO grade II). (**E**). T2WI shows a heterogeneous T2-prolonged mass in the left parietal lobe (arrow). (**F**). FLAIR shows partial signal suppression, indicating a T2-FLAIR mismatch sign (arrowheads). (**G**–**I**), T1 and T2 relaxation time and proton density (PD) maps derived from syMRI show T1 (2436 ms *) and T2 (287 ms *) relaxation time prolongations and increased PD (94.9% *) (arrows) in the tumor. * Each value is expressed as the mean. (**J**). The percentage of machine learning techniques in 90 top-five-performing models of different model categories. “Conventional MRI only” represents models developed only with conventional MRI sequences; “Advanced MRI only” for models only with advanced MRI sequences; “Mixed MRI” for models with both conventional and advanced MRI sequences; “Single sequences” for models with one sequence; “combined sequences” for models with at least two sequences; and “All sequences” for models of all sequence sets (after [60,61,64] with permission).

**Figure 8 cancers-16-01566-f008:**
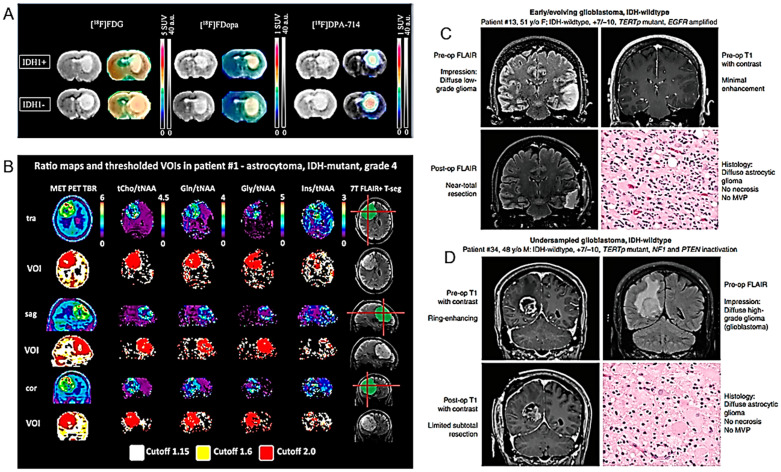
Models for the diagnosis of HGG VII. (**A**). Representative axial images of T2-w MRI and static PET imaging merged with T2-w MRI for IDH1+ and IDH1- tumors with ^18^F-FDG, ^18^F-FDopa and ^18^F-DPA-714. MRI is expressed as signal intensity and PET on a standardized uptake value (SUV) scale. (**B**). Example images of a patient with a high intratumoral correspondence between PET and MRS ratios to total N-acetylaspartate (tNAA). The actual evaluation, as performed only within the defined tumor segmentations, is shown in green. PET maps were resampled to MRS resolution. Red lines indicate slice positions. (**C**,**D**). Illustration of two representative patients highlighting the two divergent clinical scenarios where the diagnosis of “diffuse astrocytic glioma, IDH-wild-type, with molecular features of GB, WHO grade IV” based on the cIMPACT-NOW update 3, can be applied. (**C**). The first is “early/evolving” disease where the patient presents with imaging features suggestive of a lower-grade diffuse glioma (i.e., minimal to absent CE) and histology reveals a diffuse lower-grade astrocytic glioma despite extensive surgical resection. (**D**). The second is “undersampled” disease where the patient presents with imaging features of GB (i.e., ring-enhancing mass with central necrosis), but with limited surgical sampling from the infiltrative edge of the tumor, whereby histology reveals a diffuse astrocytic glioma without necrosis or microvascular proliferation (MVP) that likely would have been found upon more extensive surgical resection (after [65,73,75] with permission).

**Figure 9 cancers-16-01566-f009:**
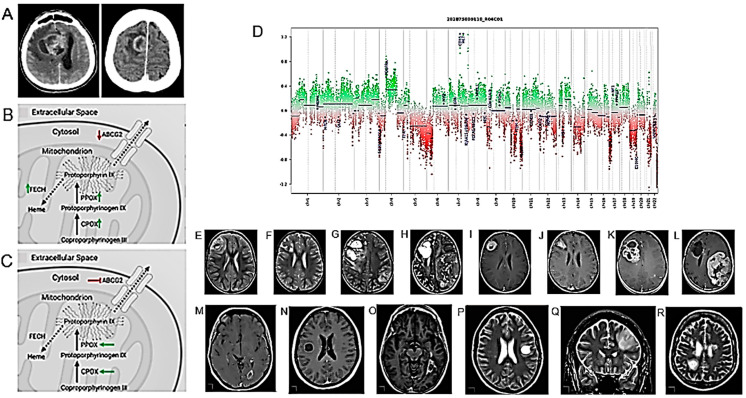
Models for the diagnosis of HGG VIII. (**A**). CT images of patients with histologically proven GB (**left**) and brain metastases (**right**). (**B**,**C**). Observed effects on intramitochondrial heme biosynthesis factors in gliomas with visible 5-ALA fluorescence and possible pharmacological targets to optimize fluorescence visualization. (**B**). Heme biosynthesis pathway activity was enhanced in gliomas with visible 5-ALA fluorescence with the upregulation of PpIX-generating factors (CPOX and PPOX; green arrow) and decreased ABCG2-mediated PpIX efflux (red arrow), outweighing the also increased further metabolization of PpIX to heme by FECH (green arrow). (**C**). Intramitochondrial heme biosynthesis factors, thus, constitute promising pharmacological targets to optimize the intraoperative 5-ALA fluorescence visualization of usually non-fluorescing tumor tissues such as LGG. In this sense, the enhancement of CPOX and PPOX (green arrows) as well as the inhibition of ABCG2 (red symbols) represent candidates for future investigations. For example, the tyrosine kinase inhibitor lapatinib constitutes a potent suppressor of ABCG2 with an overall favorable safety profile in clinical use. (**D**–**L**). Molecular profile (**D**) and MRI at four consecutive time points of an HGG infant patient. DNA methylation profile showed a complex copy number variation (CNV) with CDK6 and MET amplification, and a platelet-derived growth factor A (PDGFRA) gain (**D**). Axial TSE T2-weighted (**E**–**H**) and post-gadolinium SE T1-weighted (**I**–**L**) sequences at four consecutive time points: at presentation (**E**,**I**), at first local recurrence after surgery (**F**,**J**), at second local recurrence after reoperation (**G**,**K**) and spreading to the contralateral hemisphere (**H**,**L**). The right frontal HGG and recurrences present with a non-homogeneous hyperintense signal on T2-weighted, and intense and non-homogenous CE on T1-weighted, images (**M**). Post-contrast axial T1 MRI demonstrates close rim enhancement in a left occipital TDL with no mass effect or edema. (**N**). Post-contrast axial T1 MRI demonstrates a close regular rim enhancement pattern in a right frontal TDL with minimal mass effect and no edema. (**O**). Post-contrast axial T1 MRI demonstrates irregular closed-rim enhancement pattern in a left temporal HGG with no mass effect. (**P**). T2 axial MRI demonstrates a T2 hypointense rim in a left frontal TDL with mild edema and no mass effect. (**Q**). T2 coronal MRI demonstrates the absence of a T2 hypointense rim in the left frontal HGG. (**R**). T2 axial MRI demonstrates a T2 hypointense rim in a right partial TDL with edema and mass effect (after [79,80,84,85] with permission).

**Figure 11 cancers-16-01566-f011:**
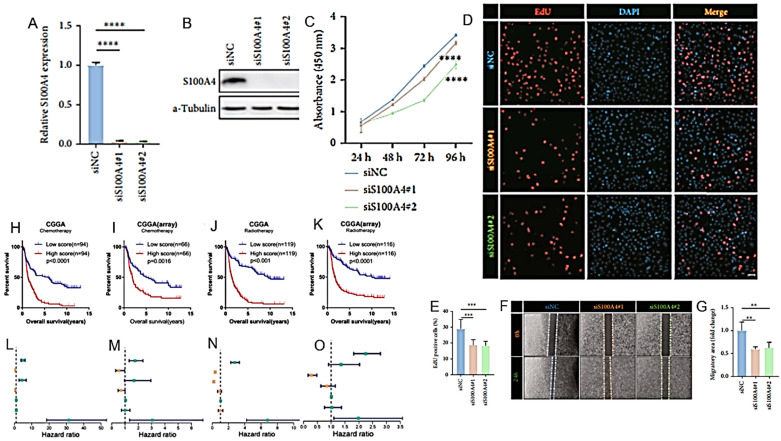
Models for the prognosis of HGG II. (**A**–**G**). S100A4 affects the proliferation and migration of glioma cells. (**A**). qRT-PCR and (**B**). Western blot analysis of S100A4 knockdown efficiency in LN229 cells. (**C**). Analysis of the proliferation of the control and S100A4-deficient LN229 cells by CCK8 assay. (**D**). Representative images and (**E**). statistical analysis of EdU assay in control and S100A4-deficient LN229 cells. (**F**). Representative images and (**G**). Statistical analysis of cell migration assay in the control and S100A4-deficient endothelial cells at the indicated times. * *p* < 0.05; ** *p* < 0.01; *** *p* < 0.001; **** *p* < 0.0001. (**H**–**O**). Angiogenesis score, as an independent prognostic factor, reflected glioma sensitivity to therapy. (**H**–**K**). Among glioma patients receiving chemotherapy (**H**,**I**) or RT (**J**,**K**), the prognosis of glioma patients in the high-score group was significantly worse than that in the low-score group in the CGGA and CGGA (array) datasets. (**L**,**O**). Univariate (**L**,**N**) and multivariate (**M**,**O**) Cox regression analysis in the TCGA (**L**,**M**) and CGGA (**N**,**O**) datasets revealed that the angiogenesis score was an independent prognostic factor for glioma patients (after [107,108] with permission).

**Figure 12 cancers-16-01566-f012:**
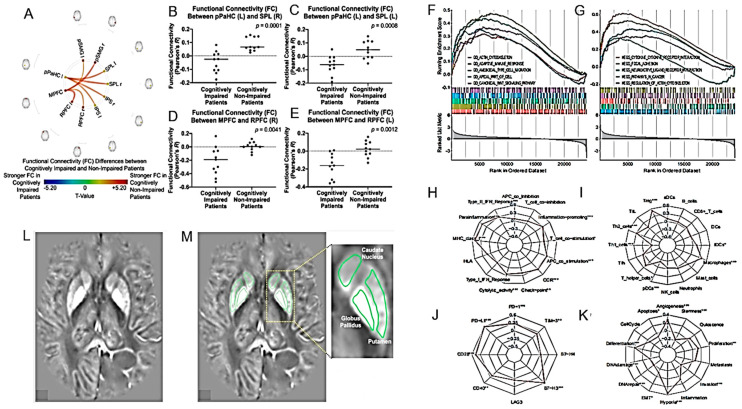
Models for the prognosis of HGG III. (**A**–**E**). Difference of ROI-to-ROI functional connectivity (FC) between cognitively impaired and non-impaired glioma patients after controlling for age. (**A**). Colors denote value of the T-statistic, yellow–red represents stronger FC in cognitively non-impaired patients, cyan-blue denotes stronger FC in cognitively impaired patients. Position of ROIs displayed on mid-axial slices. SPL = Superior Parietal Lobule; pSMG = Supramarginal Gyrus, Posterior Division; pPaHC = Parahippocampal Gyrus, Posterior Division; MPFC = Medial Prefrontal Cortex; RPFC = Rostral Prefrontal Cortex; IPS = Intraparietal Sulcus; R = Right Hemisphere; L = Left Hemisphere. (**B**–**E**) Representative comparisons of FC between cognitively impaired and nonimpaired patients, where (**B**), left pPaHC and right SPL; (**C**), left pPaHC and left SPL; (**D**), MPFC and right RPFC; and (**E**), MPFC and left RPFC. (**F**–**K**). Analysis of VEGFA expression in GB patients. (**F**), [gene ontology (GO)] and (**G**). [Kyoto encyclopedia of genes and genomes (KEGG)] analyses via gene set enrichment analysis (GSEA) between the different VEGFA expression groups. (**H**). Radar plots showing the correlation between VEGFA expression and 16 immune-related cells, (**I**). Thirteen immune-related functions, (**J**). Immune checkpoints in GB, and (**K**). Distinct functional states of cancer cells at single-cell resolution. (**L**,**M**). Basal ganglia iron levels may be used as a biomarker in glioma prognosis and treatment. (**L**). A representative quantitative susceptibility mapping (QSM) image generated with QSMnet+ and (**M**). The same QSM image with the regions of interest (ROI) outlined. An enlargement of the left basal ganglia shows the labeled ROIs for the caudate nucleus, putamen, and globus pallidus (after [110,112,113] with permission).

**Figure 13 cancers-16-01566-f013:**
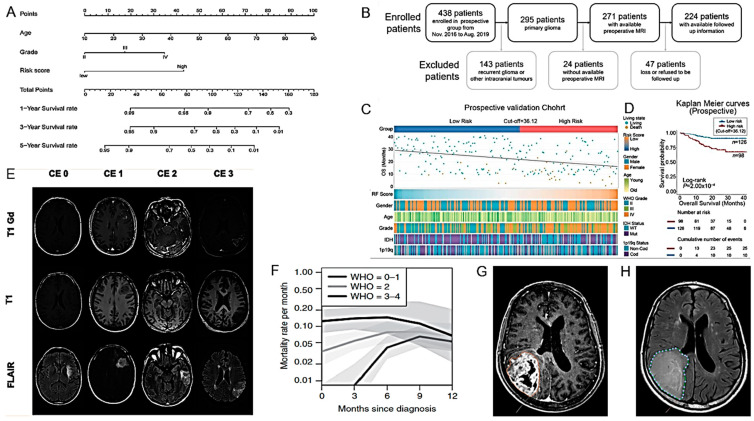
Advances in the prognosis of HGG IV. (**A**). A prognostic nomogram model using a TCGA training set with clinicopathological information for predicting the 1-, 3-, and 5-year survival rates of patients with glioma is shown. The external validation cohorts, including the CGGAseq1, CGGAseq2, and GSE16011 datasets, were used to evaluate the robustness and stability of the nomogram model. According to the results of the above univariate and multivariate Cox regression analyses, three independent prognostic indicators (age, grade, and autophagy-related risk signature) for patients with glioma were incorporated into the final prediction model. (**B**–**D**). The stability of a radiomics prediction model based on 14 features associated with immune response, especially tumor macrophage infiltration, was validated in a prospective validation cohort. (**B**). Flow diagram of glioma patients in the prospective group. A total of 224 glioma patients eligible for the study were screened from the sample of 438 glioma patients. (**C**). The heat map shows clinicopathological information of patients in different risk groups in the prospective validation cohort. (**D**). Kaplan–Meier curves show the OS of patients in the high-risk group is significantly shorter than in those in the low-risk group in the prospective validation cohort. (**E**). Patterns of CE on preoperative MRI. Patterns were defined as either: none (CE 0), patchy (multiple smaller areas of enhancement covering less than 50% of any nonenhancing tumor cross section) (CE 1), focal (1 single area of enhancement covering less than 50% of any nonenhancing tumor cross section) (CE 2), or abundant (enhancing tumor volume greater than 50% of nonenhancing tumor) (CE 3). CE, contrast enhancement. (**F**). Mortality rates of different WHO performance groups with time-dependent hazard for the groups. The mortality rates are for patients with methylated *MGMT* and resected tumor. The poorest-performing patients have significantly higher risk during the first 6–9 months, after which the better-performing patients have risk increased to a similar level. (**G**,**H**). MRI volumetric analysis of GB. (**G**). MRI in T1-weighted sequence with gadolinium showing the calculation of the lesion volume for the enhancing zone. (**H**). MRI in FLAIR-weighted sequence showing the calculation of the lesion volume for the tumor edema zone. Volumetric calculations were estimated by measuring the area of interest with a slice-by-slice semi-automatic method (after [117,118,120,122,123] with permission).

## Data Availability

The data presented in this study are available in this article.

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
