# Peer review of "Advancements in Image-Based Models for High-Grade Gliomas Might Be Accelerated"

_cancers, 2024, doi:10.3390/cancers16081566_

Round 1

Reviewer 1 Report

Comments and Suggestions for Authors

I have some comments:

The initial literature search focused on PubMed using general terms, potentially leading to the exclusion of relevant articles from other databases or using different terminology.

The criteria for inclusion/exclusion of studies were primarily based on manual assessment of article titles and abstracts, which might have introduced bias.

While preclinical studies provide valuable insights, the translation of findings to clinical practice may be limited due to differences in animal models and human pathology.

The review highlights the challenges in distinguishing between tumor tissue, pseudoprogression, and treatment-related changes using current imaging techniques. However, specific strategies for addressing these challenges are not extensively discussed.

The review acknowledges the potential of advanced imaging techniques and machine learning in improving diagnostic accuracy. However, the barriers to widespread adoption, such as standardization of protocols and variability in methodologies, are identified without proposing concrete solutions.

Author Response

The interesting comments by Reviewer 1 were dealt with as following:

  1. The initial literature search focused on PubMed using general terms, potentially leading to the exclusion of relevant articles from other databases or using different terminology.

Lines 69-72. It was pointed out that, despite the exclusion of relevant articles retrievable in other databases or using different terminology cannot be excluded, the use of the general terms indicated above has shown in the past to recover the vast majority of articles relating to the topic analyzed.   Two relevant references were introduced ([14] [15]).

  1. The criteria for inclusion/exclusion of studies were primarily based on manual assessment of article titles and abstracts, which might have introduced bias.

Lines 74-77. Similar to the above point 1, although the introduction of bias possibly originating from manual assessment of article titles and abstracts cannot be ruled out, the criteria used here for inclusion/exclusion of studies proved in the past to largely avoid this problem ([14] [15]). In case of doubt during the selection process based on the article titles, reference was made to the abstract content. In case of doubt during the selection process based on the content of the article abstracts, reference was made to the content of the whole article. In our opinion, the manual assessment of article titles and abstracts remains a most reliable procedure to select papers concerning a research topic.

  1. While preclinical studies provide valuable insights, the translation of findings to clinical practice may be limited due to differences in animal models and human pathology.

Lines 1116-1121. In agreement with Reviewer 1, we concluded that a number of novel prognostic markers for HGG have been proposed in the first semester of 2022 in preclinical or monocentric clinical studies. While preclinical studies provide valuable insights, the translation of findings to clinical practice may be limited due to differences between animal models and human pathology. Randomized multicenter clinical studies validating the prognostic capacity of most markers have not yet even started and are eagerly waited for. 

  1. The review highlights the challenges in distinguishing between tumor tissue, pseudoprogression, and treatment-related changes using current imaging techniques. However, specific strategies for addressing these challenges are not extensively discussed.

Lines 1029-1032. We pointed out that the diagnosis of HGG carried out using advanced neuroradiology techniques is still in an experimental phase and lacks standard protocols. The definition of the latter is the key to fully exploiting the potential of the many advanced imaging techniques currently available.

  1. The review acknowledges the potential of advanced imaging techniques and machine learning in improving diagnostic accuracy. However, the barriers to widespread adoption, such as standardization of protocols and variability in methodologies, are identified without proposing concrete solutions.

Lines 1032-1039. Meritorious attempts in standardization of protocols are underway by the working group of the Imaging Biomarker Standardization Initiative which has proposed standardized radiomic characteristics for phenotyping based on high-throughput images [146]. The adoption of standard algorithms should significantly improve the reproducibility of AI-based radiomics models, and represents an essential step for its widespread adoption in the clinic. Unfortunately, this is easier said than done: the definition of standardized procedures can only go through multicenter clinical studies on large samples [19] [44][147] [148].

Reviewer 2 Report

Comments and Suggestions for Authors

In this manuscript, Guido Frosina reviewed the latest progress in image-based methods for detection of high-grade gliomas. It provides a comprehensive overview of the advancement in brain tumor area, incorporating studies from MRI, PET, and preclinical animal models. While I do have a few suggestions, this review manuscript is very thorough and could benefit the potential readers in the field.

1. Please ensure that all figures are presented in high resolution. For instance, the text in Figure 7J is too small to be readable.

2. In the introduction, given the unclear causes of high-grade gliomas, I suggest shortening the discussion on concerns regarding cell phone use. This review is written for professional researchers in the field, not the general public.

3. On line 510, the author mentions that resting-state fMRI can identify vascular asynchrony in brain tumor patients. It's worth noting that another study by Hou et al. (doi.org/10.1038/s41746-023-00859-y) also demonstrated that resting-state fMRI can be used to differentiate between high-grade and low-grade tumors, as shown in Supplementary Figure 4. Incorporating this reference could enrich the discussion on the applications of fMRI in diagnosing high-grade tumor.

Author Response

The interesting comments by Reviewer 2 were dealt with as following:

In this manuscript, Guido Frosina reviewed the latest progress in image-based methods for detection of high-grade gliomas. It provides a comprehensive overview of the advancement in brain tumor area, incorporating studies from MRI, PET, and preclinical animal models. While I do have a few suggestions, this review manuscript is very thorough and could benefit the potential readers in the field.

  1. Please ensure that all figures are presented in high resolution. For instance, the text in Figure 7J is too small to be readable.

Figures 1-13. All figures have been presented at the highest resolution available in original articles. All texts have been made readable.

  1. In the introduction, given the unclear causes of high-grade gliomas, I suggest shortening the discussion on concerns regarding cell phone use. This review is written for professional researchers in the field, not the general public.

Lines 36-40. The discussion on concerns regarding cell phone use has been shortened to five lines and four ([1-4]) relevant references.

  1. On line 510, the author mentions that resting-state fMRI can identify vascular asynchrony in brain tumor patients. It's worth noting that another study by Hou et al. (doi.org/10.1038/s41746-023-00859-y) also demonstrated that resting-state fMRI can be used to differentiate between high-grade and low-grade tumors, as shown in Supplementary Figure 4. Incorporating this reference could enrich the discussion on the applications of fMRI in diagnosing high-grade tumor.

Lines 539-540. The reference by Hou et al was incorporated and discussed [116].

Reviewer 3 Report

Comments and Suggestions for Authors

- The manuscript presentes a comprehensive review of work recently published addressing research and current trends on diagnosis and prognosis of high-grade gliomas; 

- Personally, I see the proposal as a review paper, which I believe  might fit within the goals of this special issue; 

- In terms of the paper content, I would suggest:

. the conclusions to be more incisive about future research directions / clinical practice;

. a final paragraph in the introduction describing the paper structure would be helpful;

. clarifying the criterium used in selections of the further  32 paper (line 79);   

- A major concern however, is  amount of wording duplication in the manuscript from other sources, in both text and tables: iThenticate reports 45%, detailing the different origins (mostly published papers);

- Give the point above, my recommendation is for a major revision to be carried out, properly addressing text replication.

Author Response

The helpful comments by Reviewer 3 were dealt with as following:

The manuscript presents a comprehensive review of work recently published addressing research and current trends on diagnosis and prognosis of high-grade gliomas;

Personally, I see the proposal as a review paper, which I believe might fit within the goals of this special issue;

In terms of the paper content, I would suggest:

  1. . the conclusions to be more incisive about future research directions / clinical practice;

Lines 1105-1130. The conclusions were made more incisive about future research directions / clinical practice and clearer by 1) modifying their text; 2) separating them from the Discussion section; 3) grouping them under their relevant section (“Conclusions”).

  1. . a final paragraph in the introduction describing the paper structure would be helpful;

Lines 60-66. A final paragraph in the introduction describing the paper structure was introduced as following:

“In particular, magnetic resonance imaging (MRI), positron emission tomography (PET) and other diagnostics models developed in both the preclinical and clinical setting will be reviewed. Their employment for differentiating between HGG and specific pathologies or dealing with specific aspects (e.g. diagnosis of pediatric HGG) will be analyzed. Advances in prognostic models developed preclinically and clinically will follow. Eventually, we will discuss some aspects related to the end of life of HGG patients.”

  1. . clarifying the criterium used in selections of the further 32 paper (line 79);

Lines 81-84. The criterium used in selections of the further 38 papers was clarified as following:

“In addition to the 132 articles published in the first half of 2022, this review includes 38 articles published before January 1, 2022 or after June 30, 2022, due to their important relevance to the topic addressed or the topic eventually discussed on the end of life of patients, for a total of 170 references.”.

Six relevant references were introduced: [14], [15], [116], [146], [147], [148].

A major concern however, is amount of wording duplication in the manuscript from other sources, in both text and tables: iThenticate reports 45%, detailing the different origins (mostly published papers);

  1. Give the point above, my recommendation is for a major revision to be carried out, properly addressing text replication.

Tables 1,2. As discussed at submission, the novelty and significance of our paper reside in the analysis of recent (first half of 2022) progress in image-based models for high-grade gliomas, the deadliest brain tumors, and a discussion of measures to accelerate the transfer of that progress from bench to bedside.

The text of our paper (9153 words) has minimal/unavoidable (1-10%) overlap with previous publications (Encl. 1) consistently with the originality of our work.

The high (45%) duplication rate detected by the iThenticate software can be explained by the reproduction of the original legends to the figures (permissions previously submitted) and short texts in Tables 1 e 2 summarizing the main finding of each examined paper. This principle (“Fair use”) is widely accepted by publishers/journals including MDPI (previously enclosed e-mail by Ms. Talaya Zhu on Aug 19, 2021 as an instance) to avoid significantly blurring Figures and Tables understanding to the readers.

To accomplish the Reviewer 3 request to reduce as much as possible the duplication rate,  I made my best to rephrase the listed main findings in Tables 1 and 2 always prioritizing that understanding to the reader would not be significantly blurred.

Round 2

Reviewer 1 Report

Comments and Suggestions for Authors

No comments.

Reviewer 2 Report

Comments and Suggestions for Authors

The author has addressed all me comments. I recommend proceeding to publish it as it is.

Reviewer 3 Report

Comments and Suggestions for Authors

I acknowledge that in this new version of the paper, my previous comments and suggestions have been properly addressed by the author.